# Multiomics in silico analysis identifies TM4SF4 as a cell surface target in hepatocellular carcinoma

Kah Keng Wong[ID]1*, Suzina Sheikh Ab. Hamid2,3

1 Department of Immunology, School of Medical Sciences, Universiti Sains Malaysia, Kubang Kerian, Kelantan, Malaysia, 2 Tissue Bank Unit, School of Medical Sciences, Universiti Sains Malaysia, Kubang Kerian, Kelantan, Malaysia, 3 Department of Otorhinolaryngology-Head & Neck Surgery, School of Medical Sciences, Universiti Sains Malaysia, Kubang Kerian, Kelantan, Malaysia

* kahkeng@usm.my

## Abstract

The clinical application of cellular immunotherapy in hepatocellular carcinoma (HCC) is impeded by the lack of a cell surface target frequently expressed in HCC cells and with minimal presence in normal tissues to reduce on-target, off-tumor toxicity. To address this, an in silico multomics analysis was conducted to identify an optimal therapeutic target in HCC. A longlist of genes (n = 12,948) expressed in HCCs according to The Human Protein Atlas database were examined. Eight genes were shortlisted to identify one with the highest expression in HCCs, without being shed into circulation, and with restrictive expression profile in other normal human tissues. A total of eight genes were shortlisted and subsequently ranked according to the combination of their transcript and protein expression levels in HCC cases (n = 791) derived from four independent datasets. TM4SF4 was the top-ranked target with the highest expression in HCCs. TM4SF4 showed more favorable expression profile with significantly lower expression in normal human tissues but more highly expressed in HCC compared with seven other common HCC therapeutic targets. Furthermore, scRNA-seq and immunohistochemistry datasets showed that TM4SF4 was absent in immune cell populations but highly expressed in the bile duct canaliculi of hepatocytes, regions inaccessible to immune cells. In scRNA-seq dataset of HCCs, *TM4SF4* expression was positively associated with mitochondrial components and oxidative phosphorylation Gene Ontologies in HCC cells (n = 15,787 cells), suggesting its potential roles in mitochondrial-mediated oncogenic effects in HCC. Taken together, TM4SF4 is proposed as a promising cell surface target in HCC due to its high expression in HCC cells with restricted expression profile in non-cancerous tissues, and association with HCC oncogenic pathways.

## Introduction

Hepatocellular carcinoma (HCC) is the most common type of primary liver cancer where it accounts for approximately 80–90% of the cases [1,2]. Although HCC is the sixth most common cancer globally, HCC is the second leading cause of cancer-related deaths and its

**Data availability statement:** The codes used for the scRNA-seq datasets analysis are

available at: https://github.com/kahkengwong/TM4SF4_HCC_scRNA-seq

**Funding:** This work was supported by the Fundamental Research Grant Scheme (FRGS) (FRGS/1/2023/SKK10/USM/02/14 and grant no: 203.PPSP.6171388) awarded to KKW by the Ministry of Higher Education (MoHE) Malaysia. The funders had no role in study design, data collection and analysis, decision to publish, or preparation of the manuscript.

**Competing interests:** The authors have declared that no competing interests exist.

incidence is increasing worldwide [1,3,4]. In terms of mass-directed biopsies, HCC is the most frequently occurring liver lesion [5]. Majority of HCC patients are diagnosed at an advanced stage due to absence of signs and symptoms in the early stages, and the 5-year survival of HCC is dismal at 18–21% [6,7].

Approximately half of HCC patients are treated with systemic therapies, *i.e.,* the kinase inhibitors sorafenib or lenvatinib as the first-line treatment, and regorafenib (kinase inhibitor), cabozantinib (tyrosine kinase inhibitor) or ramucirumab (anti-VEGFR2 antibody) as the second-line treatment [8,9]. In the past five years, multiple immune checkpoint inhibitors (ICIs) such as atezolizumab, bevacizumab, pembrolizumab and ipilimumab, either as monotherapy or in combination with other drugs, have been approved as they enhance the overall survival (OS) of HCC patients [3,10]. Nonetheless, ICI regimens mainly confer numerical improvements in median OS compared with systemic therapies (median OS ~ 19 months vs ~ 8–14 months) [3], and only ~ 20% of HCC patients present with objective radiological responses when treated with anti-PD-1/PD-L1 antibodies [11].

Immunotherapies such as chimeric antigen receptor (CAR) T cell therapy are under investigation for HCC. However, their clinical translation is impeded by the heterogeneity of cell surface protein expression and off-tumor toxicities due to target expression in normal tissues [12]. One example is glypican-3 (GPC3; a heparan sulfate proteoglycan), where its expression is low in normal liver tissues but expressed in around 70% of HCC patients [13,14]. However, GPC3 is expressed in multiple normal tissues including the lungs, mesothelium, mammary epithelium, ovaries and the brain [15–17]. Therapeutic T cells targeting antigen expressed in normal critical tissues including the brain and lung tissues often result in toxicities [18–20]. Furthermore, serum GPC3 is commonly increased in HCC due to cell surface GPC3 shedding, and shed GPC3 (sGPC3) is a negative regulator of cell surface GPC3 leading to inhibition of HCC cell growth [21,22], and blocking anti-GPC3 CAR T cells binding, reducing treatment efficacy [23].

In this in silico study, an exhaustive exploration for the optimal cell surface target highly expressed in HCC but absent from majority of normal human tissues was conducted according to the combination of multiple publicly available transcriptomics and proteomics datasets. Potential therapeutic targets were objectively shortlisted, and TM4SF4 was identified as the most promising candidate. By leveraging multiple independent datasets, as well as comparison with GPC3 and other common surface targets studied in HCC, the therapeutic potential of TM4SF4 was established. Furthermore, the potential functional roles of TM4SF4 in HCC were explored, further supporting its future therapeutic development against the deadly malignancy.

## Materials and methods

### Datasets of the project

This is an in silico study utilizing publicly available multiomics datasets, to identify and characterize potential cell surface therapeutic targets in HCC. Inclusion criteria for datasets used in this study were: 1) Transcriptomics and proteomics datasets with open access to raw and processed data; 2) Datasets that specifically include HCC and matched control NTL group to allow for paired analysis; 3) Datasets with a sample size of at least 30 per group for quantitative comparisons; 4) Proteomics datasets utilizing mass spectrometry or similar high-throughput methods to quantify protein levels; 5) Datasets with sufficient metadata, including clear labeling of tumor and non-tumor cases, and clinical or demographic data where available. Exclusion criteria for datasets used in this study were: 1) Datasets not related to gene or protein expression, such as genomic mutation-only studies; 2) Datasets

without clear annotations of tumor or non-tumor samples; 3) Studies focusing on other liver diseases (e.g., hepatitis, cirrhosis) without sufficient HCC cases; 4) Datasets containing metastatic or recurrent HCC samples only without sufficient primary HCC cases; 5) Datasets with incomplete metadata (*e.g.*, missing labels of tumor or non-tumor cases) or lacking information about experimental protocols (*e.g.*, the specific platform used in their high-throughput analysis).

The liver cancer transcriptome comprised of 12,948 genes according to the Human Protein Atlas (HPA) database (version 20.1) [24–28] and each of the gene's features were retrieved from the database including cellular location (nuclear, cytoplasmic, or plasma membrane), transcript expression levels in normal human tissues, brain tissues and cancers, as well as concentration of the encoded protein in the blood if detected. In terms of cellular localization, the data were based on HPA's immunofluorescent staining in cell lines and UniProt cellular localization annotation. The transcript expression levels in normal human tissues were according to HPA's Consensus Dataset whereby it consisted of the combination of three transcriptomics datasets including HPA, Functional Annotation of the Mammalian Genome (FANTOM5) [29,30] and Genotype-Tissue Expression (GTEx) [31,32]. For normal human brain tissues, their transcript expression profiles were based on HPA's Consensus Human Brain Dataset, *i.e.,* combination of the human brain transcriptomics datasets of GTEx and FANTOM5. The RNA-seq data of 17 human cancer types by HPA were in accordance to The Cancer Genome Atlas (TCGA) [33,34]. Presence of encoded protein in the blood was according to the mass spectrometry dataset from the PeptideAtlas [35,36].

In addition to the aforementioned datasets, separate transcriptomics and proteomics datasets were also obtained. The transcriptomics dataset of HCC cases (n = 366) from TCGA were retrieved separately from the cBioPortal database [37,38]. The Chinese Human Proteome Project (CNHPP) RNA-seq dataset of 35 fresh paired tissues, *i.e.,* 35 adjacent non-tumor liver (NTL) cells and 35 HCC cases were retrieved from the supplementary information of its publication [39]. An independent transcriptomics dataset (microarray) of HCC (n = 225) and paired NTL (n = 220) were obtained from the Gene Expression Omnibus database (ID: GSE14520) [40,41]. For proteomics dataset, the Chinese HCC (CHCC) dataset of 166 paired tissues, *i.e.,* 166 HCC and 166 NTL cases whose proteins were measured by mass spectrometry (MS) were retrieved from the Proteomic Data Commons database (ID: PDC000198) [42]. Summary of the framework of this study is presented in Fig 1. This in silico study did not involve the direct collection of data from human participants, nor did it utilize human specimens or tissues in a manner requiring ethical approval. All data used in this study were obtained from publicly available sources and anonymized datasets. Therefore, additional review and approval by an institutional review board or ethics committee was not required for this study.

The shortlisting procedures were initiated with the list of 12,948 genes expressed in liver cancers as annotated by The Human Protein Atlas database (v20.1) before the next shortlisting steps were conducted to shortlist for genes absent from normal brain and lung tissues, restricted expression in other normal tissues but highly expressed (>50 FPKM by RNA-seq) in HCCs, and genes that potentially encode cell surface proteins without being shed into circulation. A total of eight genes were shortlisted before the ranking procedures based on the combination of four transcriptomics and proteomics HCC datasets. The top-ranked target was subsequently selected to compare its expression profile in normal human tissues and HCC cases vs GPC3 and other common therapeutic targets in HCCs. The potential functions of TM4SF4 were examined via GO enrichment analysis (in scRNA-seq dataset) and heatmap construction of genes contributing to the enriched GOs.

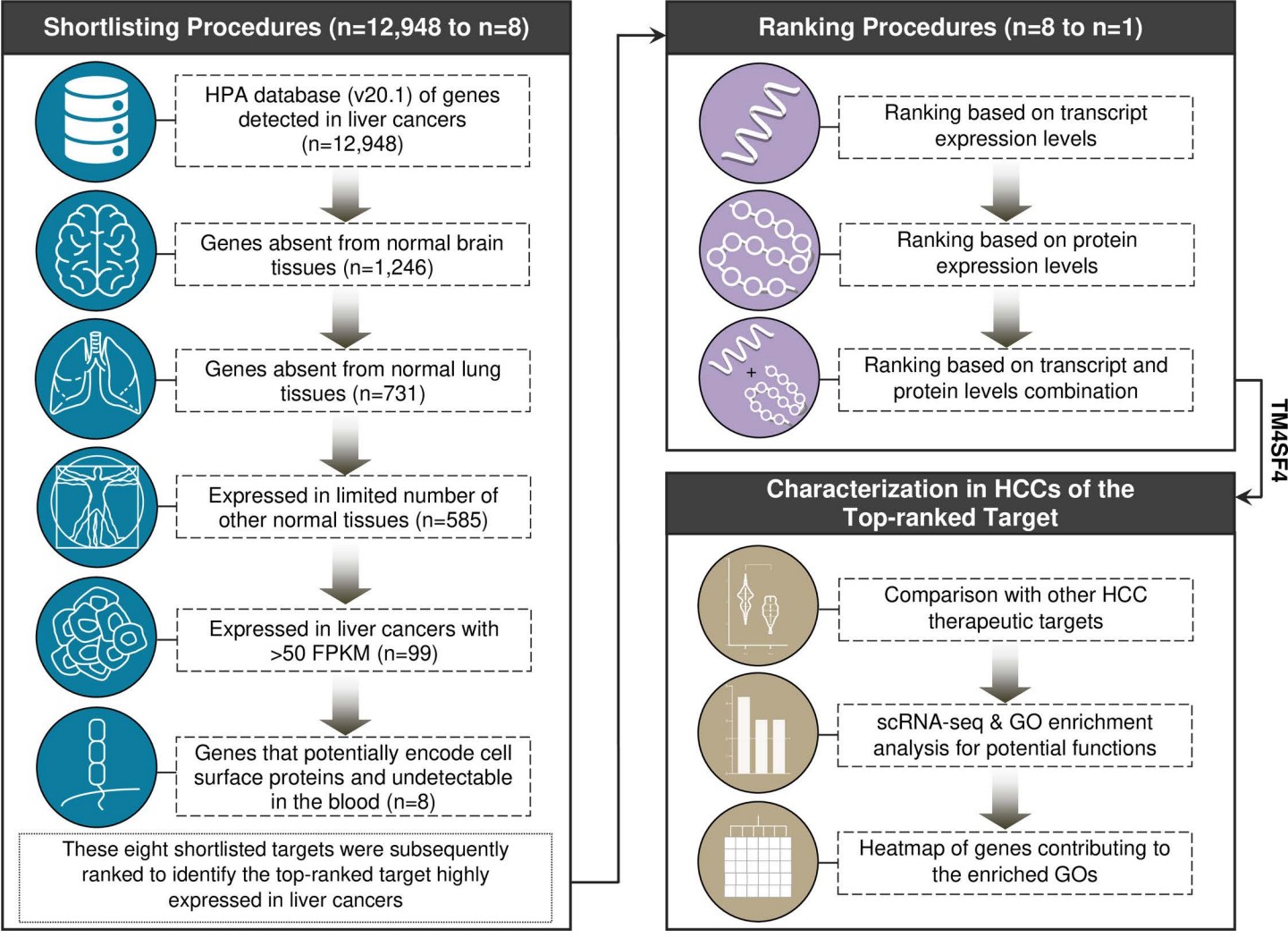

**Fig 1. Workflow to identify the optimal cell surface target in HCCs and downstream analysis to examine its potential functions in HCCs.**

## Shortlisting for surface targets with low levels of expression in normal human tissues but high levels in HCCs

The following workflow for shortlisting and ranking procedures have previously been published [43] and adopted here for HCC with modifications. Within the longlist of 12,948 genes expressed in liver cancers defined by HPA, the following sequential shortlisting steps were conducted:

(i)   **Shortlisting for genes not expressed in normal human brain tissues:** This is required as cellular immunotherapy such as the approved anti-CD19 CAR T cells have been demonstrated to cause on-target, off-tumor neurotoxicity due to the presence of CD19 expression in brain tissues [44–46]. Hence, genes with the annotation "Detected in all", "Detected in many", "Detected in some", "Detected in single" or with unknown status within the "RNA brain regional distribution" annotation were excluded. This yielded a total of 1,246 genes for the next shortlisting procedure.

(ii)  **Shortlisting for genes absent from the normal human lung tissues:** This is also required as CAR T cells therapy can cause on-target, off-tumor pulmonary toxicities [18,47,48]. From the aforementioned 1,246 shortlisted genes, those genes with expression values of above 1.5 within

"Tissue RNA - lung [NX]" ("NX" denotes normalized expression of three independent datasets, i.e., the HPA, GTEx and FANTOM5 datasets) annotation were considered as expressed in lung tissues and thus excluded. This resulted in 731 genes for the next shortlisting step.

(iii) **Shortlisting for genes with restricted expression profile in other normal human tissues:** From the aforementioned 731 shortlisted genes, those with the annotation "Detected in all" or "Detected in many" within the "RNA tissue distribution" annotation were excluded, resulting in a total of 585 genes.

(iv) **Shortlisting for genes highly expressed in HCC cases:** From the aforementioned 585 shortlisted genes, the genes annotated with > 50 FPKM for HCC within "RNA cancer specific FPKM" annotation were shortlisted, and > 50 FPKM has been adopted as the cutoff for high expression levels [49,50]. This yielded a total of 99 shortlisted genes.

(v) **Shortlisting for genes that encode cell surface proteins and not shed into circulation:** From the aforementioned 99 shortlisted genes, to select for genes that potentially encode cell surface proteins, the subcellular locations for each encoded protein were examined according to the annotation by COMPARTMENTS localization data available on GeneCards database [51,52]. A specific gene was considered to potentially encode cell surface protein according to the following criteria: (1) The "plasma membrane" annotation must appear as the top subcellular site with the highest confidence value according to the GeneCards database; (2) When a gene's top subcellular sites were annotated as "plasma membrane" and "extracellular", this gene was disregarded because it might be momentarily localized on the plasma membrane for secretion into the bloodstream; (3) Genes without any subcellular location annotation were excluded. A total of ten genes were shortlisted. The final shortlisting step involved the assessment whether each of the ten genes' encoded protein was detected in the blood by mass spectrometry based on HPA database of re-analyzed mass spectrometry dataset from the PeptideAtlas [35,36]. This yielded a final total of eight shortlisted genes and the basic features of these eight genes were summarized in S1 Table.

## Ranking of the eight shortlisted cell surface targets at transcript and protein levels

In order to shortlist for the most ideal cell surface target from the eight genes shortlisted above, the next step was to rank them in accordance with their transcript (TCGA, CNHPP, and GSE14520 datasets) or protein (CHCC dataset) expression levels (median value) in HCC cases. By comparing each gene's or protein's median expression values in HCC cases in a particular dataset, each gene or protein was given a rank score (RS). For TCGA (n = 366), GSE14520 (n = 225) and CHCC (n = 165) HCC datasets, the RS ranging from 0.1 to 0.8 was assigned for each of the eight shortlisted genes according to median expression value ranking. For example, a specific gene with the highest median transcript value compared with the rest of the eight shortlisted genes in TCGA HCC cases was assigned with the highest RS of 0.8, followed by the second, third and fourth highest median transcript levels assigned with the RS of 0.7, 0.6 and 0.5, respectively. These can be summarized as follows:

$$\text{TCG}\underline{\text{A}} \quad : \ 0.1 \leq \text{RS}_A \leq 0.8$$

$$\underline{\text{C}}\text{NHPP} : \ 0.1 \leq \text{RS}_C \leq 0.8$$

$$\underline{\text{G}}\text{SE14520} : \ 0.1 \leq \text{RS}_G \leq 0.8$$

$$\text{CH}\underline{\text{C}}\text{C} : \ 0.1 \leq \text{RS}_H \leq 0.8$$

where the underlined alphabet represents its corresponding subscripted suffix.

Then, each RS was divided by the constant value x, where x is the greatest RS value for each distinct dataset, in order to normalize the RS value to a maximum value of 1. This resulted in each dataset's normalized score (NS) that can be summarized as follows:

$$TCG\underline{A} \quad : NS_A = (RS_A)/0.8$$

$$\underline{C}NHPP : NS_C = (RS_C)/0.8$$

$$\underline{G}SE14520 : NS_G = (RS_G)/0.8$$

$$C\underline{H}CC \quad : NS_H = (RS_H)/0.8$$

For transcriptomics datasets (*i.e.,* TCGA, CNHPP, and GSE14520), each NS was then averaged to obtain the transcript mean normalized score, $\bar{x}_T$, *i.e.,* $\bar{x}_T = (NS_A + NS_C + NS_G)/3$

For proteomics dataset consisting of sole dataset (*i.e.,* CHCC), its protein mean normalized score, $\bar{x}_P$, is equivalent to $NS_H$.

Finally, both $\bar{x}_T$ and $\bar{x}_P$ were averaged in accordance with the following weighted average formula:

$$\bar{x}_{TP} = (m.\bar{x}_T + n.\bar{x}_P) / (m + n)$$

where m = the number of items in the $\bar{x}_T$ set (*i.e.,* 3; $NS_A$, $NS_C$ and $NS_G$), n = the number of item in the $\bar{x}_P$ set (*i.e.,* 1; $NS_H$).

The $\bar{x}_{TP}$ for each of the shortlisted targets represents the final score for ranking based on the combination of both transcript and protein expression levels in HCCs. Downstream characterization of the selected cell surface target (*i.e.,* TM4SF4) was conducted to determine its expression profile compared with GPC3 and other common cell surface therapeutic targets in HCC, *i.e.,* CD24 [53,54], CD133 (*PROM1*) [55,56], CD147 (*BSG*) [57,58], EPCAM [59,60], MET [61,62], and MUC1 [63,64]. Control groups were used extensively in this study. Normal human tissues as controls were included in the comparisons during target shortlisting and validation. These included normal tissues datasets from the FANTOM5 (n = 45), GTEx (n = 54), and the HPA (n = 253). Moreover, non-cancerous human tissues from the HPA immunohistochemistry (IHC) datasets (*e.g.,* gallbladder, duodenum, small intestine, stomach, pancreas, colon) were used as controls for comparison with HCC tissues in terms of TM4SF4 protein expression patterns, as detailed in subsequent sections. Additionally, NTL single cells were used as controls for intra-tumoral comparisons to identify genes specifically correlated with TM4SF4 expression in HCC but not NTL cells, in single-cell RNA-seq (scRNA-seq) analysis as detailed in subsequent sections.

## scRNA-seq data analysis

The following scRNA-seq datasets were retrieved from the Gene Expression Omnibus database: 1) Human liver scRNA-seq dataset (GSE149614) of ten HCC patients and eight adjacent NTL liver cells [65]; 2) Human gallbladder scRNA-seq dataset (GSE134355) of two non-cancerous adult individuals [66]. The initial stage of the scRNA-seq data analysis using the Seurat package [67] involved loading gene expression matrices for each dataset into R data frames, which were then transformed into sparse matrix formats. An Seurat object was subsequently created (separately for the liver and gallbladder scRNA-seq datasets),

excluding features present in less than 200 cells and filtering out cells with fewer than 500 expressed features or a total RNA count of zero. To exclude cells potentially affected by apoptosis or stress, mitochondrial gene content filtering was conducted. The percentage of mitochondrial genes per cell was calculated and cells with mitochondrial gene content above the 99th percentile were excluded from the analysis. Data normalization was then conducted using log-normalization method, the data were then scaled, and cell cycle effects were regressed out using a predefined list of S and G2M phase genes. Next, doublets were detected and removed using the scDblFinder package. To address batch effects such as variations in sequencing depth or sample handling, the datasets were integrated using Seurat package FindIntegrationAnchors function that identified integration anchors (pairs of cells from different datasets that are similar based on the specified features) between datasets. The anchors were then used by the IntegrateData function to integrate the datasets. Subsequently, Elbow plot was constructed using the ElbowPlot function to aid in determining the number of principal components that guides clusters identification using the FindNeighbors and FindClusters functions. The specific cell type for each cluster was identified according to the most enriched ontology in the Coexpression module of the ToppGene database using the top 50 genes (adjusted $p$-values) within a cluster. Dimensionality reduction was visualized using Uniform Manifold Approximation and Projection (UMAP) where the UMAP coordinates were extracted, and a data frame was created to associate each cell with its corresponding cluster identity. For visualization, the overall UMAP plots were generated with ggplot2 and colored using the viridis package. In addition, UMAP plots illustrating the expression gradient of *TM4SF4* across all cell populations were generated. The gradient ranged from light pink to dark red, representing the spectrum of *TM4SF4* expression levels using the colorRampPalette function. Cells that did not express *TM4SF4* were depicted in light gray. Additionally, a combined plot integrating both boxplot and jitter plot elements was produced to demonstrate the expression levels of *TM4SF4* across all identified cell populations using ggplot2 and colored using viridis. The codes used for the scRNA-seq datasets analysis are available at: https://github.com/kahkengwong/TM4SF4_HCC_scRNA-seq

## Gene Ontology (GO) enrichment analysis

The potential functions of TM4SF4 in HCC were investigated via GO enrichment analysis of genes highly correlated with *TM4SF4* transcript expression in HCC cases vs adjacent NTL cells. The GO enrichment analysis was conducted according to genes demonstrating high Pearson correlation value (r ≥ 6 as the cut-off) with *TM4SF4* expression in the scRNA-seq dataset GSE149614 (HCC n = 10; NTL n = 8) [65]. Cells with zero *TM4SF4* transcript count were excluded, and Pearson correlation values for each gene with *TM4SF4* expression were calculated in HCC or NTL cells separately. The set of genes demonstrating r ≥ 0.6 with *TM4SF4* expression in HCC or NTL cells were used for GO enrichment analysis according to the ToppGene database as described previously [68–73]. Corrected $p$-value, i.e., $q$-value according to the Benjamini-Hochberg procedure of < 0.01 was considered as significant enrichment.

## Statistical analysis

The Shapiro-Wilk test was conducted to examine the normality of data distribution. The unpaired t-test or Mann-Whitney test was performed to compare numerical variables between two different groups for normally and not normally distributed data, respectively. One-way ANOVA and the Holm-Sidak's multiple comparisons test were used to examine differences between more than two groups of numerical variables with normal data distribution.

Kruskal-Wallis test and the Dunn's multiple comparisons test were used to compare numerical variables in more than two groups with not normally distributed variables. For all comparisons, two-tailed $p < 0.05$ was considered as statistically significant. Statistical analyses were conducted by using GraphPad Prism version 9.0 (GraphPad Inc., La Jolla, CA, USA). The variables investigated in this study, across multiple transcriptomics and proteomics datasets, included: 1) Gene and protein expression levels in HCC vs NTL and other normal tissues (such as the brain and lung tissues); 2) Expression levels of TM4SF4 vs other common cell surface markers studied in HCC; 3) Correlation analyses (Pearson's r values) to shortlist for genes highly correlated with TM4SF4 in HCC but not NTL cells; 4) Functional annotations of these TM4SF4-correlated genes in HCC via ontologies enrichment analysis to uncover TM4SF4-specific processes in HCCs.

## Results

### TM4SF4 is the top-ranked surface target highly expressed in HCCs

A total of eight genes (*SLC2A2*, *SLC10A1*, *SLC22A7*, *SLC38A4*, *SLCO1B1*, *TFR2*, *TM4SF4*, and *TM4SF5*) were shortlisted according to the aforementioned shortlisting methodologies. Each gene was absent from normal human brain and lung tissues, exhibited a restricted transcript expression profile in other normal human tissues, highly expressed in HCCs, and potentially localized on the plasma membrane without evidence of being shed into circulation.

The expression levels of each gene in HCC cases were initially compared at the transcript level in order to identify which of the eight shortlisted genes had the highest expression in HCCs. This was performed in the TCGA (n = 366), CNHPP (n = 35) and GSE14520 (n = 225) datasets. Transmembrane 4 L six family member 4 (*TM4SF4*) was the top-ranked gene with the highest transcript expression levels in two of the datasets (*i.e.,* CNHPP and GSE14520) and it was ranked the second in the TCGA dataset according to the mean normalized score of transcript expression ($\bar{x}_T$) (Table 1). In terms of protein levels by MS in the CHCC dataset (n = 165), TM4SF5 (another member of the transmembrane 4 superfamily) demonstrated the highest expression levels followed by TM4SF4 and other proteins according to the mean normalized score of protein expression ($\bar{x}_P$) (Table 1 and S1 Fig). Finally, the weighted combination of $\bar{x}_T$ and

**Table 1. The mean normalized score (NS; derived from rank score, RS) of the eight shortlisted genes at the transcript ($\bar{x}_T$), protein ($\bar{x}_P$) or both transcript and protein ($\bar{x}_{TP}$) levels in four HCC datasets (combined total of 791 HCC cases).**

| Gene | Transcript Levels | | | | | | | Protein Levels | | Combined and weighted mean $\bar{x}_{TP}$ (transcript and protein)† (max score: 1) |
|---|---|---|---|---|---|---|---|---|---|---|
| | TCGA (n = 366) | | CNHPP (n = 35) | | GSE14520 (n = 225) | | $\bar{x}_T$ (transcript) | CHCC (n = 165) | | |
| | $RS_A$ | $NS_A$ | $RS_C$ | $NS_C$ | $RS_G$ | $NS_G$ | | $RS_H$ | $NS_H$, i.e., $\bar{x}_P$ (protein) | |
| *TM4SF4* | 0.7 | 0.875 | 0.8 | 1 | 0.8 | 1 | 0.958 | 0.7 | 0.875 | 0.938 |
| *SLC2A2* | 0.6 | 0.750 | 0.6 | 0.750 | 0.7 | 0.875 | 0.792 | 0.5 | 0.625 | 0.750 |
| *TFR2* | 0.8 | 1 | 0.7 | 0.875 | 0.2 | 0.250 | 0.708 | 0.6 | 0.750 | 0.719 |
| *TM4SF5* | 0.2 | 0.25 | 0.5 | 0.625 | 0.4 | 0.500 | 0.458 | 0.8 | 1 | 0.594 |
| *SLC38A4* | 0.5 | 0.625 | 0.3 | 0.375 | 0.6 | 0.750 | 0.580 | 0.4 | 0.500 | 0.563 |
| *SLC22A7* | 0.4 | 0.500 | 0.4 | 0.500 | 0.1 | 0.125 | 0.375 | 0.3 | 0.375 | 0.375 |
| *SLC10A1* | 0.1 | 0.125 | 0.1 | 0.125 | 0.5 | 0.625 | 0.292 | 0.2 | 0.250 | 0.281 |
| *SLCO1B1* | 0.3 | 0.375 | 0.2 | 0.250 | 0.3 | 0.375 | 0.333 | 0.1 | 0.125 | 0.281 |

†$\bar{x}_{TP} = (m.\bar{x}_T + n.\bar{x}_P) / (m+n)$.

where m = the number of items in the $\bar{x}_T$ set (*i.e.,* 3; $NS_A$, $NS_C$ and $NS_G$), n = the number of items in the $\bar{x}_P$ set (*i.e.,* 1; $NS_H$ also known as $\bar{x}_P$ due to only 1 item in this set).

protein $\bar{x}_P$, i.e., $\bar{x}_{TP}$ (combination of both transcript and protein levels ranking) was used as the consolidated ranking metric whereby TM4SF4 was the top-ranked target ($\bar{x}_{TP}$: 0.938) followed by SLC2A2 ($\bar{x}_{TP}$: 0.750), TFR2 ($\bar{x}_{TP}$: 0.719), TM4SF5 ($\bar{x}_{TP}$: 0.594) and the rest of the targets (Table 1). Hence, TM4SF4 was the selected cell surface target for downstream analysis.

## Comparison of TM4SF4 with other cell surface HCC therapeutic targets in normal human tissues and HCC cases

To validate the robustness of TM4SF4 as an optimal therapeutic target in HCC, the expression of TM4SF4 in normal human tissues and HCC cases were compared with those of seven other cell surface proteins commonly being researched for HCC therapeutic development, *i.e.,* CD24, CD133 (*PROM1*), CD147 (*BSG*), EPCAM, GPC3, MET, and MUC1. In normal human tissues, *TM4SF4* expression was significantly lower than each of the aforementioned seven genes in the FANTOM5 (n = 45 different tissues; $p < 0.05$ for each comparison), GTEx (n = 54; $p < 0.01$ for each comparison), and HPA (n = 253; $p < 1 \times 10^{-11}$ for each comparison) datasets (Fig 2A-2C). In these three datasets, *TM4SF4* transcript was highly expressed in the gastrointestinal (GI) tract and pancreas. Hence, TM4SF4 protein expression profile in these tissues was examined according to the immunohistochemistry (IHC) data by HPA in non-cancerous human tissues. The IHC by HPA was conducted using anti-TM4SF4 rabbit polyclonal antibody (HPA046430) that had been validated by orthogonal method whereby the antibody was mainly consistent with the transcript expression data across 45 human tissues by HPA.

Transcript expression levels of the genes were compared with *TM4SF4* in FANTOM5 (n = 45) (A), GTEx (n = 54) (B), and The Human Protein Atlas (n = 253) (C) normal human tissues datasets. The violin plots within each graph were arranged from left to right according to descending median expression values of the genes. Within each violin plot, the thick horizontal line represents the median, while the lower and upper horizontal lines represent the first and third quartile, respectively. Each *p*-value represents comparison of *TM4SF4* expression vs each of the other gene expression in all tissues within a specific dataset. Sm.: Small; Sig.: Sigmoid; STPM: Scaled transcript per million; TPM: Transcript per million.

In scRNA-seq of normal gallbladder (n = 3,790 cells), a total of 11 clusters were identified consisting of mesenchymal stromal cells, fibroblasts (two clusters, suggesting the presence of two subtypes of fibroblasts), neutrophils, cholangiocytes, macrophages (two clusters), NK cells, antigen presenting cells, NK and NKT cells, and endothelial cells (Fig 3A). For qualitative observations, *TM4SF4* was distinctly expressed in cholangiocytes (cluster no. 4) with sporadic low-level expression in other cell clusters (Fig 3B). Quantitative comparison showed that *TM4SF4* was significantly more highly expressed in cholangiocytes compared with all other cell clusters ($p < 0.001$) (Fig 3C). In scRNA-seq of normal liver (n = 27,246 cells), a total of 16 different clusters were identified comprising of NK cells with cytotoxic T lymphocytes (CTLs), NK and NKT cells, CTLs (three clusters), Kupffer cells (two clusters), dendritic cells, liver sinusoidal endothelial cells, hepatocytes, B cells, macrovascular endothelial cells, cholangiocytes, plasma cells, smooth muscle cells, and NKT cells (Fig 3D). Qualitatively, *TM4SF4* was almost exclusively expressed in hepatocytes (cluster no. 9) and cholangiocytes (cluster no. 13) (Fig 3E). Quantitatively, *TM4SF4* was significantly more highly expressed in either hepatocytes or cholangiocytes compared with the rest of the 14 liver cell types ($p < 0.001$) (Fig 3F). Notably, *TM4SF4* was absent or very lowly expressed in both the innate (neutrophils, NK cells, macrophages, dendritic cells) and adaptive (CTLs, B cells, plasma cells, NKT cells) immune cell populations in the normal gallbladder and liver.

(A) Eleven different cell clusters in normal gallbladder (total number of cells: 3,790). APCs: Antigen presenting cells; ECs: Endothelial cells; MSCs: Mesenchymal stromal cells; (B) Intensity of *TM4SF4* expression in normal gallbladder. The number of *TM4SF4+* cells in

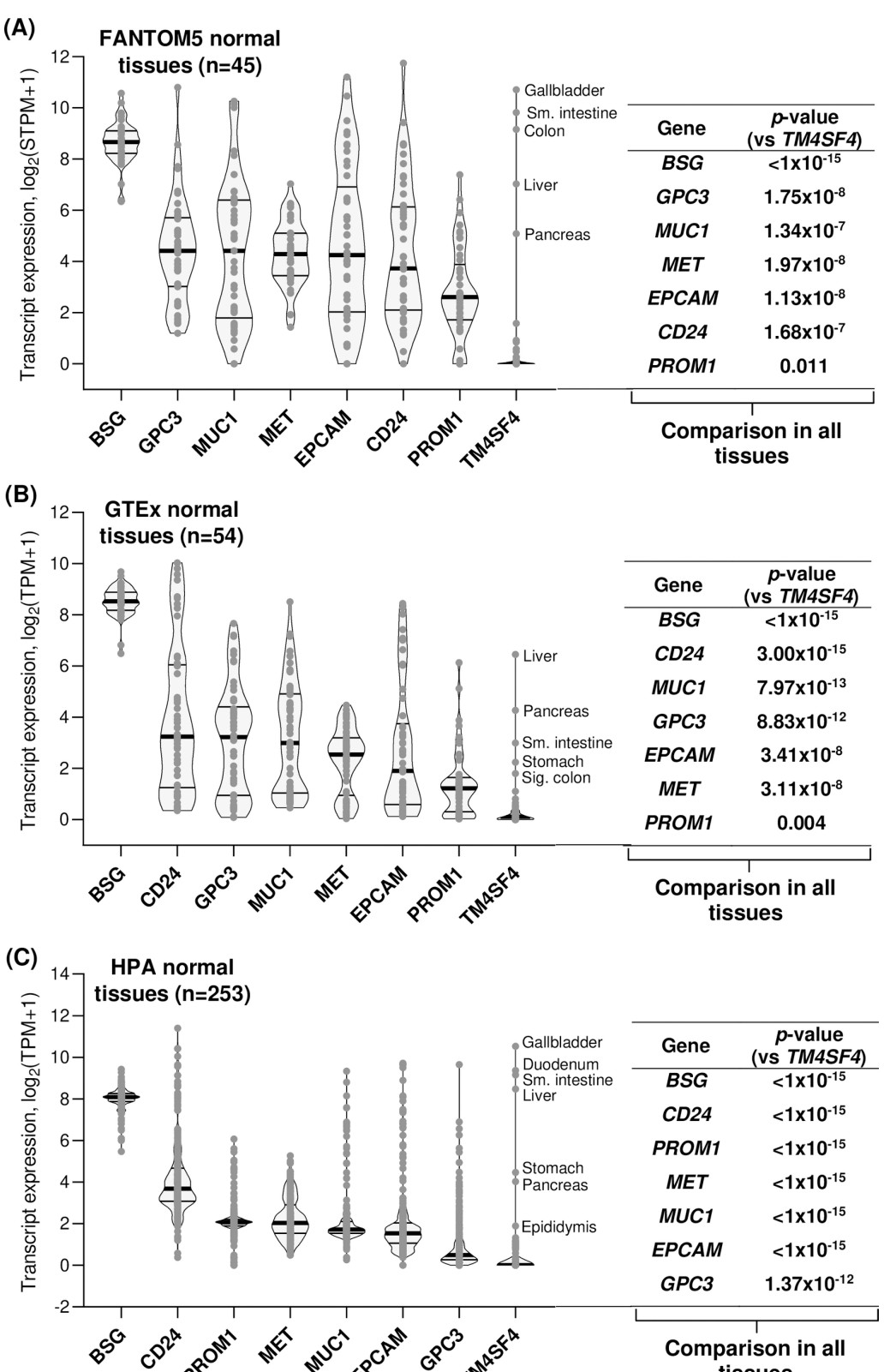

**Fig 2. Comparison of TM4SF4 expression levels with seven other therapeutic targets in normal human tissues.**

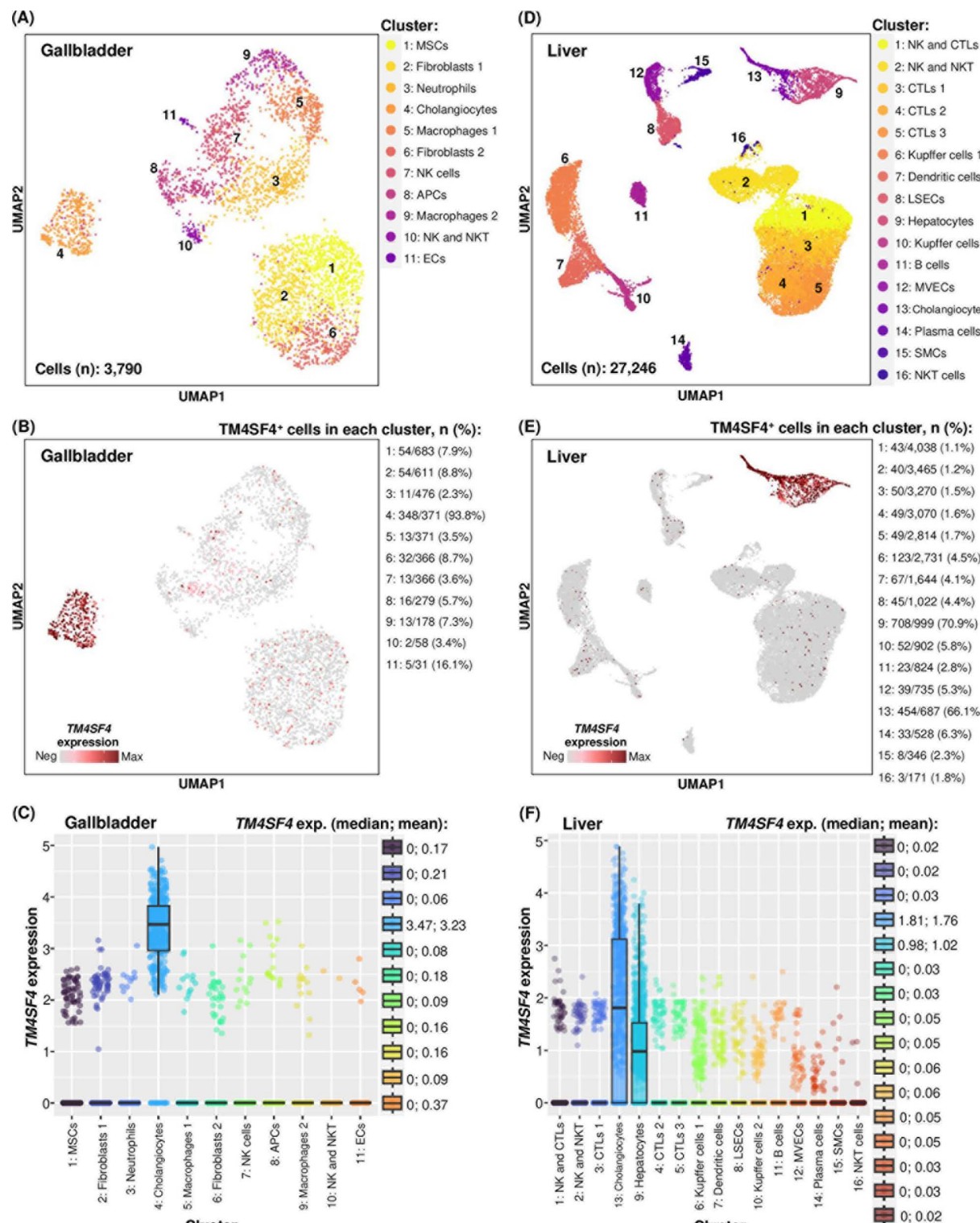

**Fig 3. Expression of *TM4SF4* in normal gallbladder and liver cells in scRNA-seq datasets.**

each cluster and the percentage are presented next to the plot; (C) Comparison of *TM4SF4* expression in cholangiocytes versus the rest of the cell clusters ($p < 0.001$ for all comparisons). *TM4SF4* expression levels in median and mean are presented next to the plot; (D) Sixteen different cell clusters in normal liver (total number of cells: 27,246). CTLs: Cytotoxic T lymphocytes; LSECs: Liver sinusoidal endothelial cells; MVECs: Macrovascular endothelial cells; SMCs: Smooth muscle cells; (E) Intensity of *TM4SF4* expression in normal liver; (F) Comparison of *TM4SF4* expression in cholangiocytes or hepatocytes versus the rest of the cell clusters ($p < 0.001$ for all comparisons).

At the protein levels, in non-cancerous gallbladder, TM4SF4 protein was strongly expressed in glandular cells with membranous staining on all sides of the plasma membrane domains (*i.e.,* apical, lateral and basal) where the basal domain was in contact with adjacent cells (*i.e.,* the lamina propria consisting of lymphocytes, loose connective tissues and blood vessels) (Fig 4). In contrast, TM4SF4 protein was highly expressed on the apical domain of enterocytes plasma membrane (*i.e.,* the microvilli of enterocytes) in the duodenum and small intestines without contact with the adjacent lamina propria. TM4SF4 protein expression was absent in the lamina propria, lymphocytes, Goblet cells, Paneth cells, intestinal stem cells,

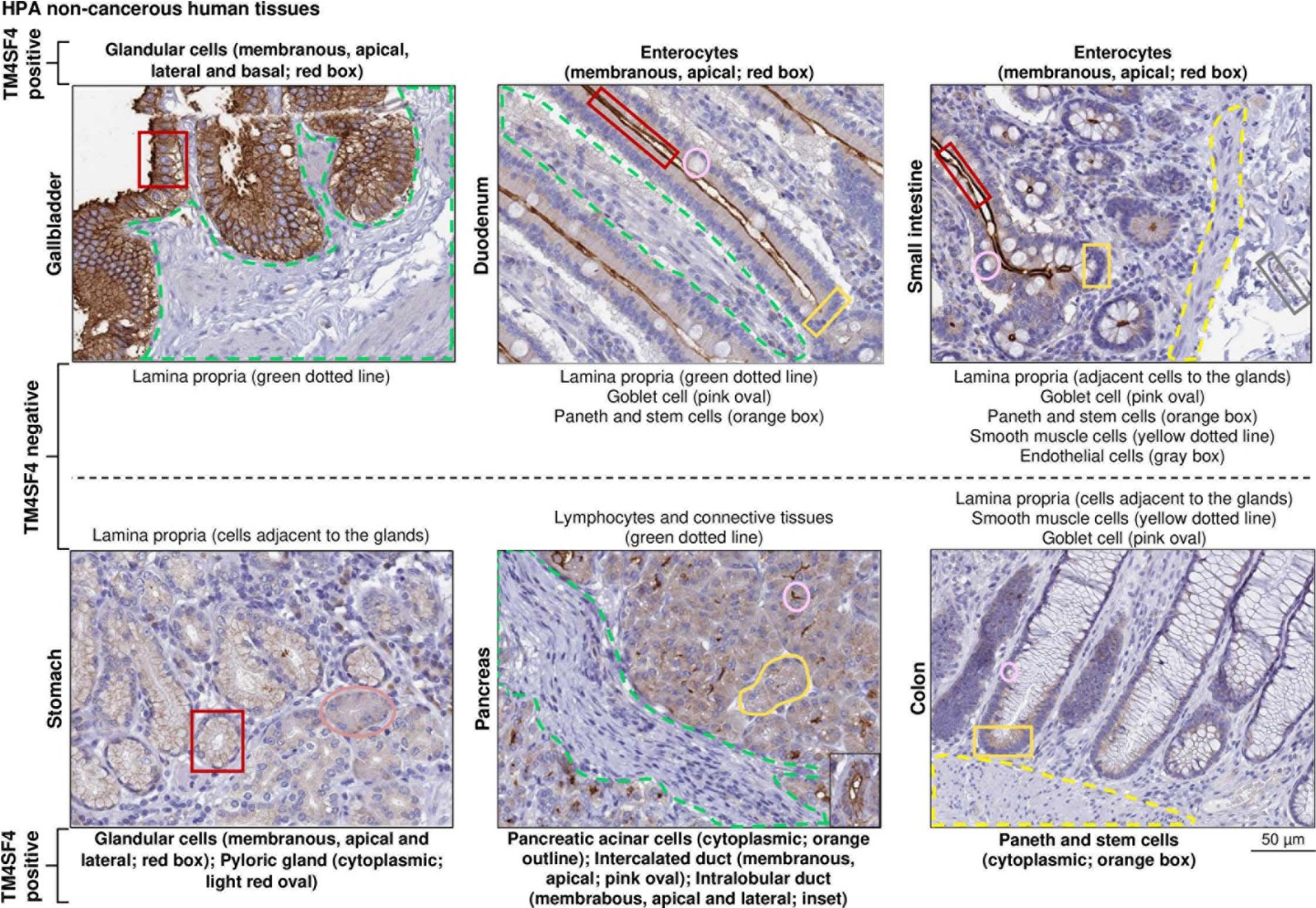

**Fig 4. TM4SF4 protein expression by IHC in GI tract and pancreas obtained from The Human Protein Atlas database.**

smooth muscle cells, and endothelial cells. Membranous TM4SF4 was also present in stomach's glandular cells but on the apical and lateral domains of plasma membrane, pancreatic intercalated ducts (apical), and intralobular ducts (apical and lateral) without contact with the lamina propria (Fig 4).

Cell populations positive or negative for TM4SF4 were labeled with boxes, outline or dotted lines. The cell populations positive for TM4SF4 were stated as membranous or cytoplasmic. For membranous staining, the plasma membrane domain demonstrating TM4SF4 positivity is also stated, *i.e.,* apical, lateral and/or basal. Next, TM4SF4 expression in HCC cases were compared with the aforementioned seven HCC therapeutic targets in three independent transcriptomics datasets, *i.e.,* TCGA (n = 366), GSE14520 (n = 225) and CNHPP (n = 35), as well as in the proteomics CHCC (n = 165) dataset. *TM4SF4* expression was significantly higher than six of the genes in TCGA dataset ($p < 1 \times 10^{-6}$ for each comparison) (Fig 5A), four of the genes in CNHPP dataset ($p < 1 \times 10^{-5}$ for each comparison; *CD24* was not present in the platform) (Fig 5B), all seven genes in GSE14520 dataset ($p < 0.01$ for each comparison) (Fig 5C), and four of the proteins in CHCC proteomics dataset ($p < 0.05$ for each comparison; CD24 was not present in the platform) (Fig 5D).

Transcript expression levels of the genes were compared with *TM4SF4* in TCGA (n = 366) (A), CNHPP (n = 35) (B), and GSE14520 (n = 225) (C) datasets, while comparison at protein levels was conducted in the CHCC (n = 165) (D) dataset. The violin plots within each graph were arranged from left to right according to ascending median expression values of the genes

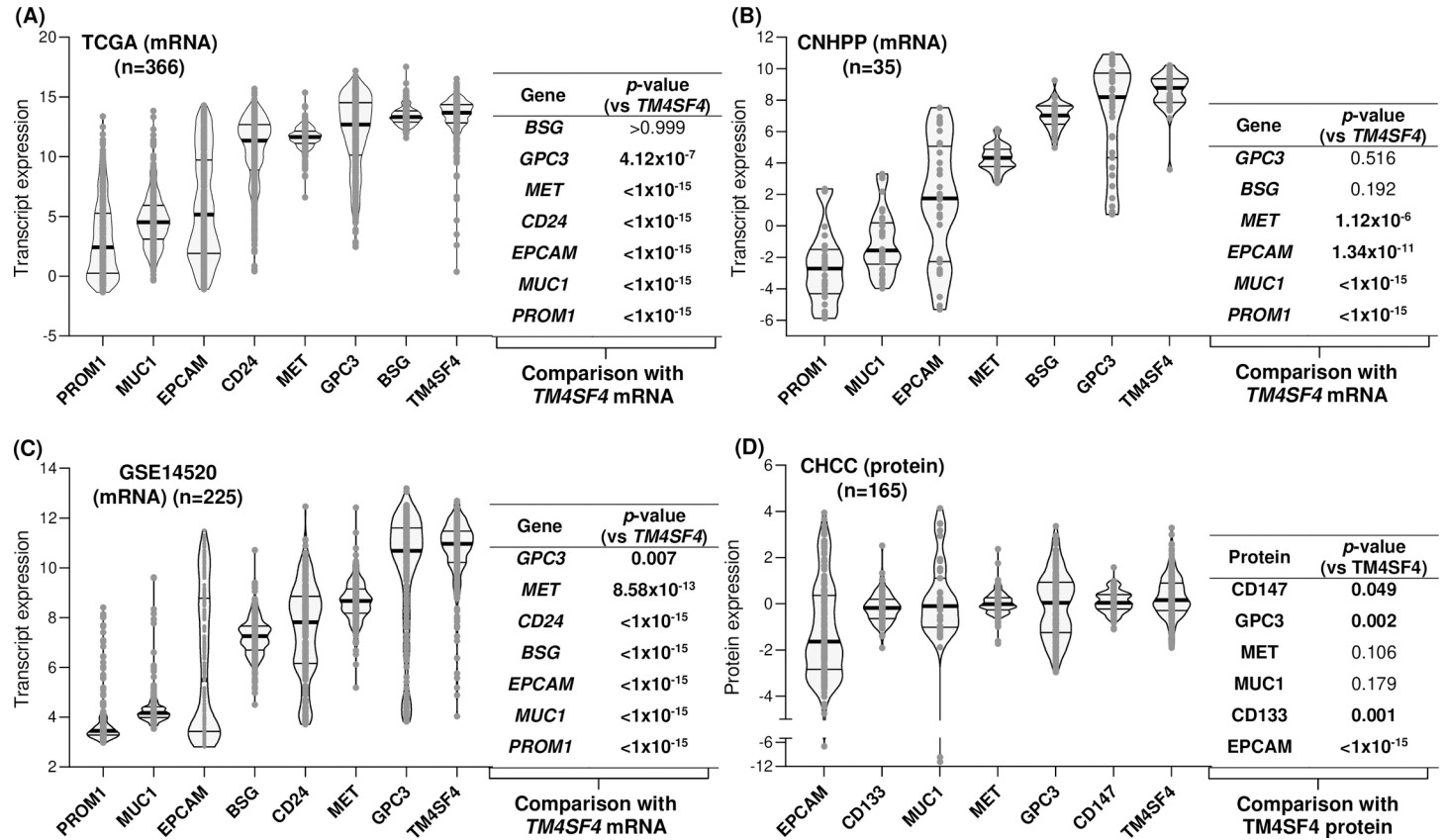

**Fig 5. Comparison of TM4SF4 expression levels with seven other therapeutic targets in HCC cases.**

or proteins. Within each violin plot, the thick horizontal line represents the median, while the lower and upper horizontal lines represent the first and third quartile, respectively. Each *p*-value represents comparison of *TM4SF4* expression vs each of the other gene expression in all HCC cases within a specific dataset.

## The relationship of TM4SF4 with clinico-demographical characteristics and survival of HCC patients

Two datasets contained comparable clinico-demographical and survival data of the HCC patients, GSE14520 (HCC n = 221) and CHCC (HCC n = 159), hence they were utilized to examine the relationship of TM4SF4 expression with those characteristics at the transcript and protein levels, respectively. TM4SF4 transcript and protein levels were not associated with the clinico-demographical characteristics examined including age, gender, cirrhosis, AFP, and ALT, except lower *TM4SF4* transcript expression in stage III-IV compared with stage I-II patients ($p = 0.014$) but this was not significant at the protein levels ($p = 0.477$) where median TM4SF4 protein levels were higher in stage III-IV than stage I-II HCC patients (S2 Fig). In terms of survival, TM4SF4 expression was not associated with OS or relapse-free survival (RFS) at both transcript and protein levels (S3 Fig).

## TM4SF4 expression is higher in HCC compared with NTL cases

Three datasets contained matched NTL cases, *i.e.,* CNHPP (n = 35 for each HCC and NTL group), GSE14520 (NTL n = 220; HCC n = 225), and CHCC (n = 166 for each HCC and NTL group), and hence they were utilized for comparison of TM4SF4 expression between both groups. *TM4SF4* expression was significantly higher in HCC cases compared with NTL cases in both CNHPP ($p = 2.58 \times 10^{-9}$) (Fig 6A) and GSE14520 ($p = 3.82 \times 10^{-9}$) (Fig 6B) datasets. Consistent with these observations, TM4SF4 protein was also more highly expressed in HCCs compared with NTLs in the CHCC proteomics dataset ($p = 1.16 \times 10^{-11}$) (Fig 6C).

Comparison of *TM4SF4* expression in NTL vs HCC cases in transcriptomics (A-C) datasets with paired control NTL cases. Comparison of TM4SF4 protein expression by IHC in NTL (D) and HCC (E) cases obtained from The Human Protein Atlas database. The frequency (in percentage) and intensity (moderate or strong) of TM4SF4 staining in HCC cells were stated with each HCC case ID. Yellow arrow: Cholangiocyte; Green arrow: Endothelial cell; Red arrow: Bile duct canaliculi; Pink arrow: Kupffer cell; Orange arrow: Hepatic stellate cell.

In terms of TM4SF4 protein levels by IHC, the NTL (n = 3) and HCC (n = 7) cases immunostained for TM4SF4 were retrieved from the HPA database. In the three available NTL cases, TM4SF4 protein was strongly expressed on the plasma membrane domain of hepatocytes that form bile duct canaliculi, while demonstrating weak cytoplasmic staining in hepatocytes (Fig 6D). TM4SF4 was also strongly expressed in cholangiocytes with membranous (all sides of the plasma membrane domain) and cytoplasmic staining. TM4SF4 expression was absent in hepatic stellate cells, Kupffer cells, and endothelial cells (Fig 6D). In HCC cases, TM4SF4 was expressed in all seven cases whereby membranous TM4SF4 expression in HCC cells on all sides of the plasma membrane domains were present in 85.7% (n = 6/7) of the HCC cases, in accordance with HPA's interpretation of "Distinct membranous immunoreactivity was observed in liver and pancreatic cancers". Three representative HCC cases with membranous TM4SF4 expression are shown in Fig 6E. A schematic diagram illustrating bile duct canaliculi positive for TM4SF4 staining and their surrounding structures in NTL is presented in Fig 7.

Immune cells such as lymphocytes and macrophages that arrive from the hepatic portal vein and hepatic artery patrol the surrounding liver cells such as hepatocytes, cholangiocytes

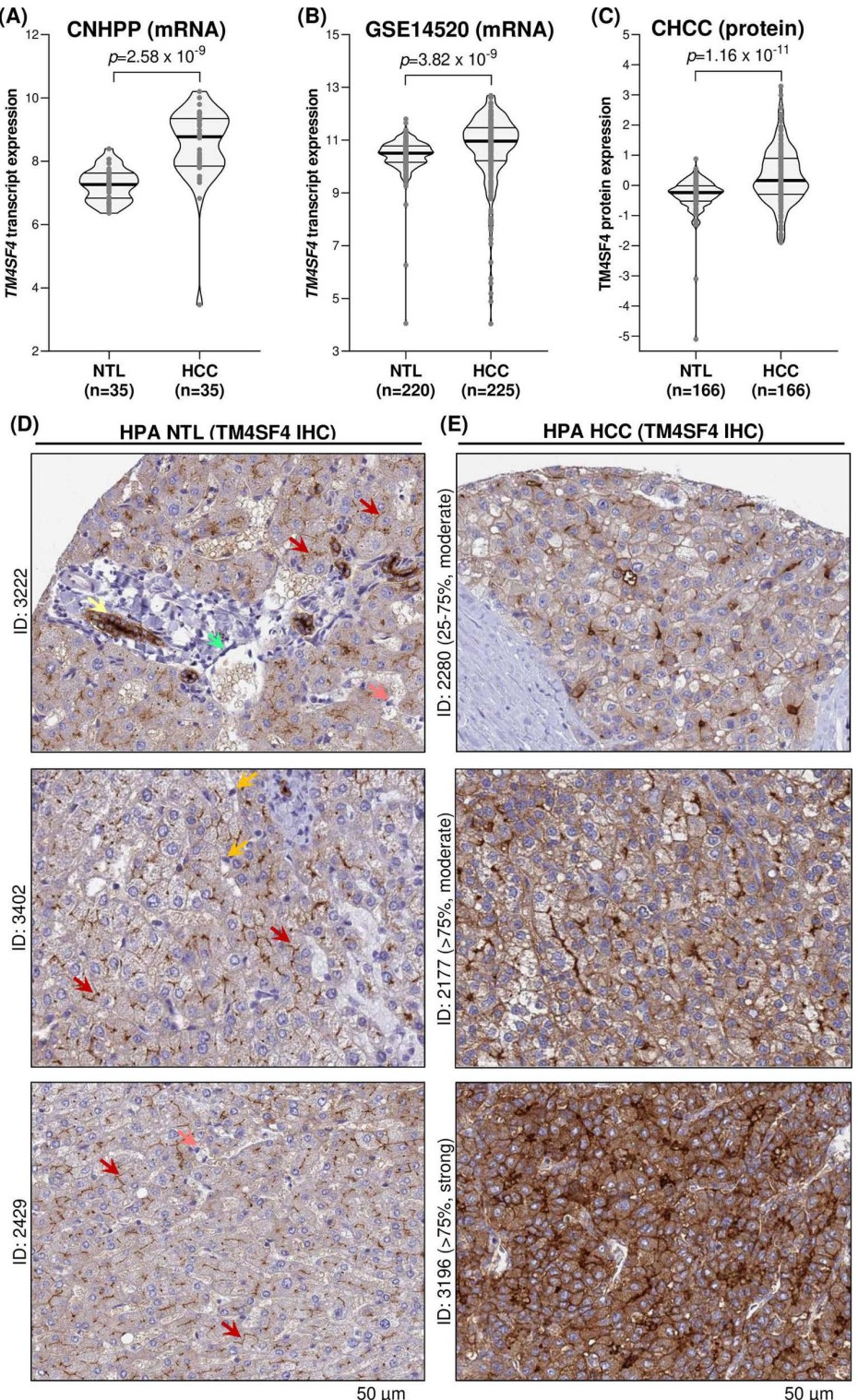

**Fig 6. TM4SF4 expression in NTL compared with HCC cases at transcript and protein levels.**

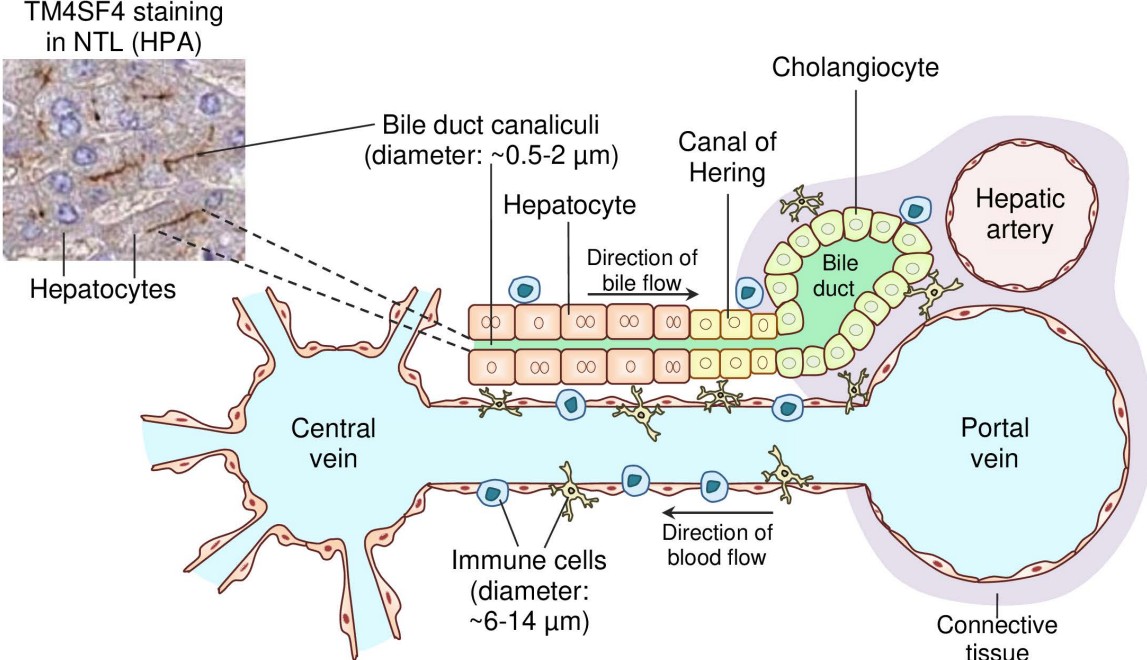

**Fig 7. Schematic diagram depicting the bile duct canaliculi and their surrounding structures in normal liver tissues according to Tam *et al.* [74].**

and the canal of Hering that are partially lined by hepatocytes and cholangiocytes. TM4SF4 is expressed on the surface of hepatocytes that form the bile duct canaliculi that are not exposed to immune cells. Meanwhile, TM4SF4 is expressed on the surface of cholangiocytes, potentially exposing them to immunotherapy. The inset TM4SF4 IHC staining image is an NTL case obtained from HPA (patient ID: 2429).

## TM4SF4 expression is associated with mitochondrial components and oxidative phosphorylation GOs in HCC cells

To gain insights on the potential functions of TM4SF4 in HCCs, GO enrichment analysis was conducted in the scRNA-seq dataset (GSE149614) of HCC cells (n = 15,787 cells derived from ten HCC patients) and matched, adjacent NTL cells (n = 1,932 cells from eight control individuals). The scRNA-seq dataset was utilized to avoid bulk transcriptome datasets that may contain mixture of different cells apart from HCC cells, *e.g.,* cholangiocytes, immune cells and/or endothelial cells.

A total of 49 and 32 distinct genes were highly correlated (*i.e.,* Pearson r > 0.6) with *TM4SF4* expression in NTLs and HCCs, respectively (S2 Table). Subsequently, GO enrichment analysis was conducted for these two sets of genes separately. In NTLs, five groups of GOs were significantly enriched (*q* < 0.01) mainly pertaining to plasma membrane components, cellular adhesion, myosin complexes, and secretory vesicles (Fig 8). In contrast, seven groups of GOs were significantly enriched (*q* < 0.01) for HCCs whereby four of the seven GO groups comprised of mitochondrial components and energy production GOs, and the other three GO groups consisted of cellular detoxification, lipid processes, and nucleotide synthesis or metabolism GOs (Fig 8). The complete list of enriched GOs and the list of genes contributing to each GO enrichment in NTLs or HCCs are presented in S3 Table. The top eight genes

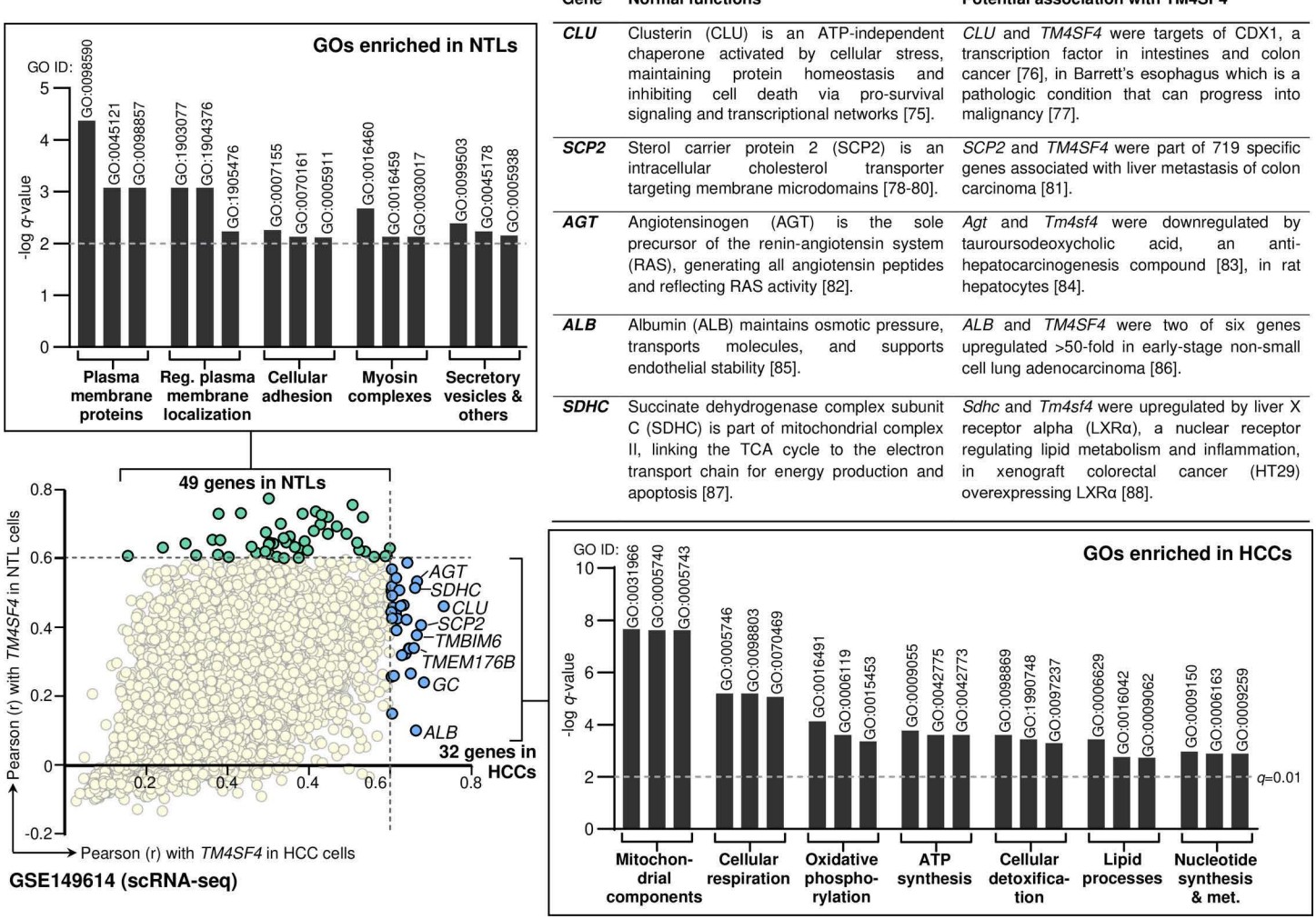

**Fig 8. GO enrichment analysis according to the genes with Pearson correlation value (r) of >0.6 with *TM4SF4* expression in NTL cells (n = 49 genes) or HCC cells (n = 32 genes).**

with the highest r values with *TM4SF4* expression are shown in the scatter plot (Fig 8). Five of these top genes (*CLU*, *SCP2*, *AGT*, *ALB*, and *SDHC*) were described in terms of their normal functions and potential association with *TM4SF4*, and such descriptions for all 32 genes are presented in S4 Table.

The enrichment analysis was conducted in the scRNA-seq dataset (GSE149614) of HCC cells (n = 15,787 cells derived from ten HCC patients) and adjacent NTL cells (n = 1,932 cells from eight HCC patients). The top eight genes with the highest r values with *TM4SF4* expression are shown in the scatter plot. The bar graphs correspond to the groups of enriched GOs, and three representative GOs (the GO ID is displayed above each bar graph) that formed each GO group labeled below the graph are demonstrated. The bar graph was drawn on the -log *q*-value scale, and the dotted horizontal line represents the *q*-value cut-off of 0.01 to be considered as significant enrichment. Notes: One gene with r > 0.6 (*i.e., ANPEP*) in HCCs was not included in the HCCs GO enrichment analysis due to it exhibited the lowest r value that satisfied the 0.6 cut-off in HCCs, and its r value was higher in NTLs than HCCs (0.630 vs 0.602).

It was thus included in the NTLs GO enrichment analysis only to avoid any overlapping gene for the analysis enriched specifically in NTLs or HCCs.

In terms of TCGA cases, each gene's expression values (transcript per million, TPM) in HCC cases (n = 369) vs non-HCC liver cases (n = 160) were compared according to the Gene Expression Profiling Interactive Analysis 2 (GEPIA2) database [75]. Six of the 32 genes (*MPC2*, *NDUFC2*, *COX6A1*, *TMCO1*, *COPZ1*, and *COX8A*), as well as *TM4SF4*, showed significantly higher expression values in HCC compared with non-HCC liver cases (all *p* < 0.001; S4 Table). The rest of the genes (n = 26) did not show significant difference between the two groups, however 22 of these genes showed higher expression values in the HCC group (S4 Table).

Finally, heatmap of the 32 genes highly correlated with *TM4SF4* expression in HCCs as well as the GO groups containing these genes are presented in Fig 9. In particular, 17 of the 32 genes (53.1%) contributed to the enrichment of mitochondrial components and energy production GOs (*i.e., CLU, SCP2, TMBIM6, SDHC, ACAA2, AMBP, MPC2, NDUFC2, NDUFB2, GSTK1, MGST3, COX6A1, PRDX6, COX8A, CFH, PHYH*, and *ATP5A1*).

This was according to the expression of the 32 genes positively associated with *TM4SF4* expression (*i.e.,* r > 0.6) as shown in the bar graph on the left of the heatmap. The number(s) in brackets for each gene corresponds to the seven GO groups (defined below the heatmap) where a specific gene contributed to their enrichment in HCCs. Genes without such number denote absence from contributing to the enrichment of any GO.

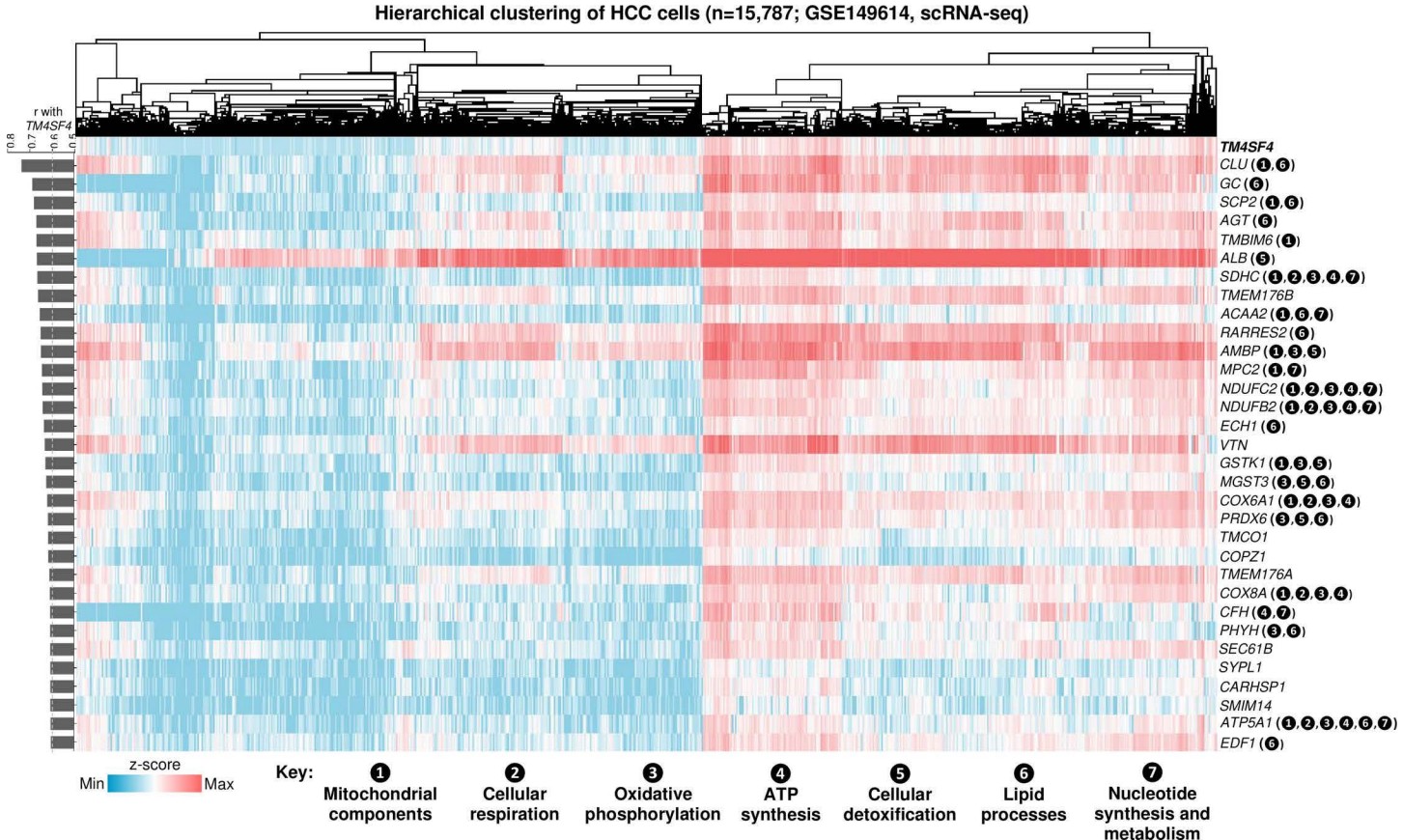

**Fig 9. Heatmap and hierarchical clustering of HCC cells (n = 15,787 cells from ten HCC patients; GSE149614).**

## Discussion

In this in silico study, through a sequence of impartial shortlisting and ranking methodologies, TM4SF4 was shortlisted and finally selected as an optimal cell surface therapeutic target for HCCs. Similar framework had previously been devised and implemented in other types of cancers, including colorectal cancer (CRC) and diffuse large B cell lymphoma, to identify key cell surface targets in these malignancies [43], and transcriptomics as well as proteomics datasets have regularly been utilized to build algorithms for uncovering crucial biomarkers in cancers [76,77]. TM4SF4 is not expressed in the critical tissues for survival including the brain, lung, heart and kidney tissues, while overexpressed in HCCs compared with matched and adjacent NTL cells. TM4SF4 also compares more favorably than other common HCC therapeutic targets in terms of its expression profile, *i.e.,* lower in normal human tissues but high in HCC cases.

The tetraspanin superfamily comprises of around 33 proteins localized in the plasma membrane where each protein consists of four transmembrane domains, short intracellular amino and carboxy tails, a small intracellular loop, and two extracellular loops [78,79]. Tetraspanins exert diverse biological functions such as cell adhesion, motility, invasion and signal transduction through their unique abilities to associate with other proteins (*e.g.,* integrins and receptor tyrosine kinases) to form tetraspanin-enriched microdomains [79,80]. The TM4SFs is a branch of the tetraspanin superfamily consisting of TM4SF1, TM4SF4, TM4SF5, TM4SF18, TM4SF19 and TM4SF20 where they significantly differ with other tetraspanins structurally [81,82]. TM4SF4, originally identified in human intestine epithelium and liver tissues [83], is specifically expressed in the gastrointestinal tract and pancreas. In the intestines, TM4SF4 plays physiological roles in the absorption and uptake of thiamine by interacting with the human thiamine transporter-2 [84], and in the liver, TM4SF4 is required for regenerating liver cells through regulation of cellular proliferation [85]. In the pancreas, TM4SF4 is highly expressed in pancreatic α cells, and the protein is used as a cell surface marker to differentiate pancreatic α from β cells [86,87].

In the past decade, TM4SF4 has been implicated in promoting the oncogenic processes of HCCs. TM4SF4 protein expression was significantly upregulated in HCC tissues compared with paired non-cancerous liver tissues [88], comparable with the observations in this study that TM4SF4 protein was upregulated in the CHCC proteomics dataset of HCCs vs paired NTLs. Functionally, TM4SF4 overexpression in HCC cell lines (QGY-7701 and BEL-7404) induced cell growth and colony formation of the cells, and TM4SF4 downregulation led to inhibited HCC cells proliferation [88]. In a separate report, TM4SF4 protein was also overexpressed in HCCs compared with paired non-tumor tissues by utilizing an in-house anti-TM4SF4 polyclonal antibody. TM4SF4 protein was specifically localized on all sides of the plasma membrane domains of HCCs but only present in the bile duct canaliculi of normal liver tissues, consistent with the IHC data by HPA as reported in this study. Knockdown of TM4SF4 in HCC cells impaired growth potential in vitro and repressed tumor growth in a mouse xenograft model of human HCC [89], similar with another study showing attenuated proliferation and migration of TM4SF4-silenced HCC cells [90]. Collectively, these studies indicate that TM4SF4 functions as an oncoprotein in HCCs.

Accumulating evidence has also established the roles of TM4SF4 as an oncoprotein in other solid tumors. In CRC, higher TM4SF4 expression is associated with liver metastasis, worse survival, tumor development and epithelial-mesenchymal transition (EMT) program regulated by TGFβ/Snail, TNFα/NFκB, and thymidylate synthase pathways [91–93]. In lung cancer, TM4SF4 expression is upregulated where it promotes tumor growth, migration, and invasion through activation of PI3K/AKT and JAK2/STAT3 pathways, as well as preserving

cancer stem cell features [94–97]. TM4SF4 is also overexpressed in pancreatic cancers as a hub gene and upregulated in chemoresistant ovarian cancer cells [98,99]. Taken together, TM4SF4 exerts oncogenic effects in multiple types of solid tumors, supporting its potential as a therapeutic target beyond HCC.

Mitochondria are the key organelles that generate energy for cellular processes via the production of adenosine triphosphate (ATP). ATP is mainly produced by oxidative phosphorylation which takes place on the inner mitochondrial membrane of the cell [100–102]. Liver cells are remarkably rich in mitochondria to support their numerous metabolic functions whereby each hepatocyte contains approximately 1,000–2,000 mitochondria [103,104], and deregulated mitochondria signaling could lead to liver pathologies including HCCs. Mitochondrial oxidative phosphorylation defects cause misfolded proteins to accumulate in the mitochondrial matrix, causing increased expression of oncoproteins that fuel HCCs development and metastasis [105].

In this study, *TM4SF4* expression was positively associated with mitochondrial components, oxidative phosphorylation, and ATP synthesis GOs. Mitochondria, often found in close proximity to the PM for $Ca^{2+}$ signaling or to provide ATP, play key roles in liver metabolism and lipid hormone responses [106–108]. In HCC, the expression of mitochondrial citrate transporter and PM citrate transporter is elevated, and inhibitors targeting these transporters induce apoptosis of HCC cells [109]. More importantly, TM4SF4 overexpression accelerates carbon tetrachloride ($CCl_4$)-mediated liver injury, and $CCl_4$-mediated liver injury is caused by mitochondrial permeability alterations [110]. Collectively, as TM4SF4 is found on the PM of HCCs, the potential interaction of TM4SF4 with mitochondria in HCC cells may lead to oncogenic effects such as mitochondrial abnormalities, but this proposal remains to be validated experimentally.

Four of the genes that contributed to the enrichment of mitochondrial components (*MPC2*, *NDUFC2*, *COX6A1*, and *COX8A*) were significantly overexpressed in TCGA HCC cases as described previously in S4 Table. As *TM4SF4* was also overexpressed in TCGA HCC cases, this suggests that TM4SF4 signaling may lead to upregulation of any of these four genes' expression. Although there have not been studies showing the direct association between TM4SF4 with these four genes, co-expression patterns in independent studies may support this observation. In particular, *MPC2* may appear as a stronger candidate among the other genes where the expression of both *MPC2* and *TM4SF4* were suppressed in human malignant epithelial cells infected by SARS-CoV-2 versus controls [111,112]. However, copy-number variation analyses showed that MPC2 was frequently amplified in a specific molecular subtype of HCC termed as the "mixed group" subtype characterized by higher variability in metabolic activity, compared to the quiescent subtype defined by low metabolic activity and cholesterol subtype characterized by reliance on cholesterol synthesis pathways [113]. Hence, direct mechanistic studies of TM4SF4 regulation of genes expression are required to demonstrate the downstream signaling controlled by TM4SF4 in HCC.

On the other hand, *TM4SF4* expression was utilized as a marker for normal cholangiocytes to separate them from normal hepatocytes, fibroblasts, lymphocytes, and endothelial cells in liver tissues based on recent scRNA-seq data [114] that tally with the scRNA-seq observations in this study. TM4SF4 protein is highly expressed in normal cholangiocytes that form the hepatic bile ducts, while absent from multiple other cell types in the liver. Although TM4SF4 protein is expressed in hepatocytes, it is localized on the surface of these cells that form the bile duct canaliculi that are not in contact with infiltrating lymphocytes and other immune cells. As such, immunotherapy such as CAR T cells targeting TM4SF4 is proposed to have a lower likelihood in causing significant toxicities on normal hepatocytes. Likewise, TM4SF4

expression in the intestinal enterocytes' apical surface (specifically, the microvilli of enterocytes lining the intestinal lumen) are unlikely to be affected by anti-TM4SF4 immunotherapy due to absence of contact with the lamina propria where immune cells including lymphocytes patrol. TM4SF4 highly expressed on the surface of HCC cells are not necessarily tumor-associated antigens due to lack of actual immunotherapeutic agent demonstrating potency against TM4SF4 in HCCs.

The limitations of this study are acknowledged as follows: 1) TM4SF4 is present in normal bile ducts (*i.e.,* cholangiocytes) and the gallbladder epithelial cells where they are in contact with the adjacent stroma containing immune cells and lymphocytes. Hence, anti-TM4SF4 immunotherapy may cause toxicity to these sites. However, immune-related injury of bile ducts after immune-checkpoint inhibitor treatment can be treated with immunosuppressive agents such as prednisolone or mycophenolate mofetil [115,116]; 2) Targeting TM4SF4 systematically can cause toxicity to pancreatic α cells that express TM4SF4 [117,118]. However, our findings have demonstrated that TM4SF4 expression is more restricted compared with other more commonly investigated HCC surface therapeutic targets (*e.g.*, GPC3 and CD147) which are highly expressed in tissues critical for survival such as the lungs and brain. Essentially, the localization of TM4SF4 in the bile duct canaliculi of the hepatocytes where immune cells are unable to access, suggests a lower likelihood of off-tumor effects when anti-TM4SF4 therapies are locally administered near HCC tumor sites; 3) The potential roles of TM4SF4 in mitochondrial-mediated oncogenic effects in HCCs require experimental verification.

## Conclusions

This study provides evidence to support the development of anti-TM4SF4 immuno-therapies such as CAR T cells against HCCs. The more restricted expression profile of TM4SF4 in normal human tissues versus other common therapeutic targets in HCC further supports the use of anti-TM4SF4 targeted agents against the disease. Moreover, the association of *TM4SF4* expression with mitochondrial processes and energy production GOs in scRNA-seq dataset of HCC cells are reported for the first time. Finally, the shortlisting and ranking methodologies proposed in this study can be implemented independently to uncover the optimal cell surface targets for therapeutic development in other malignancies.

## Supporting Information

**S1 Fig. Transcript expression levels of the eight shortlisted targets in HCC cases.** Violin plots of the transcript expression levels in TCGA (n = 366) (A), CNHPP (n = 35) and GSE14520 (n = 225) HCC cases, and the protein expression levels in CHCC (n = 165) HCC cases (D). The violin plots within each graph were arranged from left to right according to ascending median transcript or protein expression values for each gene. For each violin plot, the thick black bar in the center denotes the median while the upper and lower lines denote the third and first quartile, respectively.
(PDF)

**S2 Fig. The relationship of TM4SF4 transcript and protein expression with clinico-demographic characteristics of HCC patients.** (A) Association of TM4SF4 transcript levels with clinico-demographic characteristics of HCC patients in the GSE14520 dataset (n = 221); (B) Association of TM4SF4 protein levels with clinico-demographic characteristics of HCC patients in the CHCC dataset (n = 159).
(PDF)

**S3 Fig. The relationship of TM4SF4 transcript and protein expression with the survival of HCC patients.** (A) Association of TM4SF4 transcript levels with the survival of HCC patients in the GSE14520 dataset (n = 221); (B) Association of TM4SF4 protein levels with the survival of HCC patients in the CHCC dataset (n = 159).
(PDF)

**S1 Table. Characteristics of the eight shortlisted genes highly expressed in HCC cases but with restricted expression profiles in normal human tissues according to The Human Protein Atlas dataset.**
(XLSX)

**S2 Table. The complete list of genes and their Pearson r values with *TM4SF4* expression in NTL or HCC cells in scRNA-seq dataset (GSE149614).**
(XLSX)

**S3 Table. The list of enriched GOs (q < 0.01) according to the genes with Pearson r > 0.6 with TM4SF4 expression in HCC cells (n = 15,787 cells derived from ten HCC patients) or paired NTL (n = 1,932 cells from eight HCC patients) in scRNA-seq dataset (GSE149614).**
(XLSX)

**S4 Table. Normal functions and potential association with TM4SF4 of the genes highly correlated with TM4SF4 expression in HCC cells (n = 32).**
(DOCX)

## Acknowledgments

The authors wish to acknowledge the School of Medical Sciences and the Research Creativity and Management Office (RCMO), Universiti Sains Malaysia, for their invaluable support.

## Author contributions

**Conceptualization:** Kah Keng Wong.

**Data curation:** Kah Keng Wong.

**Formal analysis:** Kah Keng Wong.

**Funding acquisition:** Kah Keng Wong, Suzina Sheikh Ab. Hamid.

**Investigation:** Kah Keng Wong.

**Methodology:** Kah Keng Wong.

**Project administration:** Kah Keng Wong, Suzina Sheikh Ab. Hamid.

**Resources:** Kah Keng Wong, Suzina Sheikh Ab. Hamid.

**Software:** Kah Keng Wong.

**Supervision:** Kah Keng Wong.

**Validation:** Kah Keng Wong.

**Visualization:** Kah Keng Wong.

**Writing – original draft:** Kah Keng Wong.

**Writing – review & editing:** Kah Keng Wong, Suzina Sheikh Ab. Hamid.

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
