## [Decision Letter · Decision Letter 0]

29 Sep 2024

PONE-D-24-26145Multiomics data analysis identifies TM4SF4 as a cell surface target in hepatocellular carcinomaPLOS ONE

Dear Dr. Wong,

Thank you for submitting your manuscript to PLOS ONE. After careful consideration, we feel that it has merit but does not fully meet PLOS ONE’s publication criteria as it currently stands. Therefore, we invite you to submit a revised version of the manuscript that addresses the points raised during the review process.Along with addressing the reviewer’s comments, please ensure to address the following points:

The conclusion presented at the end of the abstract of manuscript should be carefully refined to ensure consistency with the final conclusion of the manuscript.In particular, it is important to address whether the identification of both TM4SF4+ and TM4SF4- cell populations across various normal human tissues impacts the validity of using TM4SF4 as a cell surface target for hepatocellular carcinoma treatment. Does the presence of TM4SF4 in non-cancerous tissues undermine its potential as a specific therapeutic target for hepatocellular carcinoma? This key consideration should be thoroughly discussed to ensure the conclusion reflects the broader implications of the findings.

Please submit your revised manuscript by Nov 13 2024 11:59PM. If you will need more time than this to complete your revisions, please reply to this message or contact the journal office at plosone@plos.org . Please include the following items when submitting your revised manuscript:

We look forward to receiving your revised manuscript.

Kind regards,

Mahmood S Choudhery, PhD

Academic Editor

PLOS ONE

Journal Requirements:

This work was supported by the Fundamental Research Grant Scheme (FRGS) (grant no: 203.PPSP.6171388) awarded to KKW by the Ministry of Higher Education (MoHE) Malaysia.”

Additional Editor Comments:

The conclusion presented at the end of the abstract of manuscript should be carefully refined to ensure consistency with the final conclusion of the manuscript. In particular, it is important to address whether the identification of both TM4SF4+ and TM4SF4- cell populations across various normal human tissues impacts the validity of using TM4SF4 as a cell surface target for hepatocellular carcinoma treatment. Does the presence of TM4SF4 in non-cancerous tissues undermine its potential as a specific therapeutic target for hepatocellular carcinoma? This key consideration should be thoroughly discussed to ensure the conclusion reflects the broader implications of the findings

Reviewers' comments:

Reviewer's Responses to Questions

**Comments to the Author**

1. Is the manuscript technically sound, and do the data support the conclusions?

Reviewer #1: Partly

Reviewer #2: Yes

2. Has the statistical analysis been performed appropriately and rigorously? 

Reviewer #1: Yes

Reviewer #2: Yes

3. Have the authors made all data underlying the findings in their manuscript fully available?

Reviewer #1: Yes

Reviewer #2: Yes

4. Is the manuscript presented in an intelligible fashion and written in standard English?

Reviewer #1: Yes

Reviewer #2: Yes

5. Review Comments to the Author

Reviewer #1: This paper presents a detailed analysis of TM4SF4, whose expression is increased in HCC, by using various databases. In addition, immunostaining with anti-TM4SF4 antibodies were performed to confirm positive cells in HCC tissues. As noted in the conclusion, TM4SF4-neutralizing antibodies may be a new therapeutic target in HCC. However, the correlation between TM4SF4 overexpression and poor prognosis was low from the KM plot (Fig. S3). This suggests that TM4SF4 is not directly associated with tumor growth, and that genes with correlated high expression or interacting downstream genes or signaling pathway may be most important.

(1) Reanalyze the scRNA-seq data in Fig. 9, and check whether the representative genes that correlate with TM4SF4 expression are consistent with the proteins that have been published in Gene Card and other journals.

Are these genes related to TM4SF4 associated with HCC malignancy by searching such as TCGA database?

I would like to conduct additional analysis of the above, and hope that the impact of TM4SF4 on downstream signals will lead to the meaningful to develop an anticancer drug for targeting TM4SF4.

(2) Please explain why the authors showed the result in Fig.8? Although the structure of the inhibitory site of TM4SF4 protein is abruptly introduced in Fig. 8, I do not feel the significance of treating it in the text unless the inhibitors are actually synthesized and tested in mice or other animals.

Reviewer #2: The authors of this article have applied a multi-omics data analysis of TM4SF4 as a cell surface target in hepatocellular carcinoma. This title is worthy of investigation and it is evident that lots of effort was put into writing this informative manuscript. However, some comments are mentioned below that need more clarification, explanation, or modification.

1. Please include the type of study in the title.

2. The abstract of the article is better to be structured.

3. The third paragraph of the introduction section needs to be concise. The amount of details that were provided may distract the attention of the reader from the main purpose of the study.

4. The authors have discussed some methods of the study in the last paragraph of the introduction section. Let them just remain in their appropriate place, i.e. the method section. You should include two things at the end of the introduction section: 1. the objective(s) of the study and 2. the novelty of the study

5. Please mention the type of the study in the method section.

6. Have you utilized any control group or not?

7. Have you provided any ethics approval number from a recognized institution? If yes, please mention it.

8. It is recommended to transfer all the website links to the references section.

9. Please mention the inclusion and exclusion criteria for the cases in your study.

10. Please mention the variables that were investigated in the methods section.

11. Please improve the cohesion and coherence of the discussion section and make it concise. It does not needed to report the findings of every previous study in a separate sentence. For instance, it is better to combine the findings of several similar previous studies and present them in one informative sentence in order to avoid redundancy.

Wish you luck and prosperity in revising the manuscript.

6. PLOS authors have the option to publish the peer review history of their article (what does this mean? ). If published, this will include your full peer review and any attached files.

**Do you want your identity to be public for this peer review?** For information about this choice, including consent withdrawal, please see our Privacy Policy .

Reviewer #1: **Yes: ** Sadahiro Iwabuchi

Reviewer #2: **Yes: ** Ali Kashkooe

---

## [Author Response · Author response to Decision Letter 1]

21 Dec 2024

Academic Editor

The conclusion presented at the end of the abstract of manuscript should be carefully refined to ensure consistency with the final conclusion of the manuscript.

Thank you for the comments. The conclusion at the end of the Abstract has been revised as follows:

Taken together, TM4SF4 is proposed as a promising cell surface target in HCC due to its high expression in HCC cells with restricted expression profile in non-cancerous tissues, and association with HCC oncogenic pathways.

In particular, it is important to address whether the identification of both TM4SF4+ and TM4SF4- cell populations across various normal human tissues impacts the validity of using TM4SF4 as a cell surface target for hepatocellular carcinoma treatment. Does the presence of TM4SF4 in non-cancerous tissues undermine its potential as a specific therapeutic target for hepatocellular carcinoma? This key consideration should be thoroughly discussed to ensure the conclusion reflects the broader implications of the findings.

Thank you for the comments. The issues raised (i.e., whether TM4SF4 expression in non-cancerous tissues undermine its therapeutic potential) have now been addressed in the limitations section of the Discussion as follows (pages 26-27):

The limitations of this study are acknowledged as follows: 1) TM4SF4 is present in normal bile ducts (i.e., cholangiocytes) and the gallbladder epithelial cells where they are in contact with the adjacent stroma containing immune cells and lymphocytes. Hence, anti-TM4SF4 immunotherapy may cause toxicity to these sites. However, immune-related injury of bile ducts after immune-checkpoint inhibitor treatment can be treated with immunosuppressive agents such as prednisolone or mycophenolate mofetil [129, 130]; 2) Targeting TM4SF4 systematically can cause toxicity to pancreatic α cells that express TM4SF4 [131, 132]. However, our findings have demonstrated that TM4SF4 expression is more restricted compared with other more commonly investigated HCC surface therapeutic targets (e.g., GPC3 and CD147) which are highly expressed in tissues critical for survival such as the lungs and brain. Essentially, the localization of TM4SF4 in the bile duct canaliculi of the hepatocytes where immune cells are unable to access, suggests a lower likelihood of off-tumor effects when anti-TM4SF4 therapies are locally administered near HCC tumor sites. **********

The Conclusion section of the manuscript has also been revised as follows (page 27):

This study provides evidence to support the development of anti-TM4SF4 immunotherapies such as CAR T cells against HCCs. The more restricted expression profile of TM4SF4 in normal human tissues versus other common therapeutic targets in HCC further supports the use of anti-TM4SF4 targeted agents against the disease. Moreover, the association of TM4SF4 expression with mitochondrial processes and energy production GOs in scRNA-seq dataset of HCC cells are reported for the first time. Finally, the shortlisting and ranking methodologies proposed in this study can be implemented independently to uncover the optimal cell surface targets for therapeutic development in other malignancies.

References

129. Haanen J, Carbonnel F, Robert C, Kerr KM, Peters S, Larkin J, et al. Management of toxicities from immunotherapy: ESMO Clinical Practice Guidelines for diagnosis, treatment and follow-up. Ann Oncol. 2017;28(suppl_4):iv119-iv42. doi: 10.1093/annonc/mdx225. PMID: 28881921.

130. Murayama A, Tajiri K, Nakaya A, Ito H, Hayashi Y, Entani T, et al. Intrahepatic Bile Duct Injury as a Hepatic Immune-Related Adverse Event after Immune-Checkpoint Inhibitor Treatment. Case Rep Gastroenterol. 2021;15(2):645-51. doi: 10.1159/000516199. PMID: 34616270.

131. Wendt A, Eliasson L. Pancreatic alpha-cells - The unsung heroes in islet function. Semin Cell Dev Biol. 2020;103:41-50. doi: 10.1016/j.semcdb.2020.01.006. PMID: 31983511.

132. Roder PV, Wu B, Liu Y, Han W. Pancreatic regulation of glucose homeostasis. Exp Mol Med. 2016;48(3):e219. doi: 10.1038/emm.2016.6. PMID: 26964835.

Journal’s Editorial

Thank you for the comments. We have implemented all the requirements as necessary.

This work was supported by the Fundamental Research Grant Scheme (FRGS) (grant no: 203.PPSP.6171388) awarded to KKW by the Ministry of Higher Education (MoHE) Malaysia.”

Thank you for the comments. The funder had no role in the study apart from providing the research funding, thus the statement “The funders had no role in study design, data collection and analysis, decision to publish, or preparation of the manuscript.” is correct.

Thank you for the comments. The reference list has been checked and it is complete and correct.

Reviewer #1

This paper presents a detailed analysis of TM4SF4, whose expression is increased in HCC, by using various databases. In addition, immunostaining with anti-TM4SF4 antibodies were performed to confirm positive cells in HCC tissues. As noted in the conclusion, TM4SF4-neutralizing antibodies may be a new therapeutic target in HCC. However, the correlation between TM4SF4 overexpression and poor prognosis was low from the KM plot (Fig. S3). This suggests that TM4SF4 is not directly associated with tumor growth, and that genes with correlated high expression or interacting downstream genes or signaling pathway may be most important.

(1) Reanalyze the scRNA-seq data in Fig. 9, and check whether the representative genes that correlate with TM4SF4 expression are consistent with the proteins that have been published in Gene Card and other journals.

Are these genes related to TM4SF4 associated with HCC malignancy by searching such as TCGA database?

I would like to conduct additional analysis of the above, and hope that the impact of TM4SF4 on downstream signals will lead to the meaningful to develop an anticancer drug for targeting TM4SF4.

Thank you for the comments. For the original version of the manuscript’s Fig. 9 (now designated as Fig. 8 due to the exclusion of a main figure, i.e., the molecular docking figure previously designated as Fig. 8, to address the second comment below), we have conducted literature review on all 32 genes associated with TM4SF4 expression in HCC but not in NTLs (according to r >0.6 cut-off). There has not been a functional study demonstrating a direct link between TM4SF4 with any of those 32 genes due to the lack of transcriptomics- or proteomics-wide study upon knockdown or overexpression of TM4SF4 expression in HCC cells that may alter the expression of these genes.

Nonetheless, potential associations between TM4SF4 with any of the 32 genes are shown by shared altered expression profile in specific malignancies or conditions. For instance, SCP2 and TM4SF4 were part of 719 specific genes associated with liver metastasis of colon carcinoma [81], or Agt and Tm4sf4 were downregulated by tauroursodeoxycholic acid, an anti-hepatocarcinogenesis compound [83], in rat hepatocytes [84].

We have included a new table in Fig. 8 summarizing the potential links of five selected top genes most highly associated with TM4SF4, as well as each gene’s normal functions for added contexts. The complete descriptions the normal functions of all 32 genes and their potential links with TM4SF4 are presented in a new S5 Table. We have also labeled the top eight genes (selected the top eight due to space constraint in the figure) within the scatter plot of Fig. 8 for better clarity.

The following descriptions have been added in the Results section (page 20):

The top eight genes with the highest r values with TM4SF4 expression are shown in the scatter plot (Fig. 8). Five of these top genes (CLU, SCP2, AGT, ALB, and SDHC) were described in terms of their normal functions and potential association with TM4SF4, and such descriptions for all 32 genes are presented in S5 Table.

Regarding TCGA cases, the following descriptions have been added in the Results section (page 21):

In terms of The Cancer Genome Atlas (TCGA) cases, each gene's expression values (transcript per million, TPM) in HCC cases (n=369) vs non-HCC liver cases (n=160) were compared according to the Gene Expression Profiling Interactive Analysis 2 (GEPIA2) database [89]. Six of the 32 genes (MPC2, NDUFC2, COX6A1, TMCO1, COPZ1, and COX8A), as well as TM4SF4, showed significantly higher expression values in HCC compared with non-HCC liver cases (all p<0.001; S5 Table). The rest of the genes (n=26) did not show significant difference between the two groups, however 22 of these genes showed higher expression values in the HCC group (S5 Table).

Regarding potential TM4SF4 downstream signaling, the following descriptions have been added in the Discussion section (page 25):

Four of the genes that contributed to the enrichment of mitochondrial components (MPC2, NDUFC2, COX6A1, and COX8A) were significantly overexpressed in TCGA HCC cases as described previously in S5 Table. As TM4SF4 was also overexpressed in TCGA HCC cases, this suggests that TM4SF4 signaling may lead to upregulation of any of these four genes’ expression. Although there have not been studies showing the direct association between TM4SF4 with these four genes, co-expression patterns in independent studies may support this observation. In particular, MPC2 may appear as a stronger candidate among the other genes where the expression of both MPC2 and TM4SF4 were suppressed in human malignant epithelial cells infected by SARS-CoV-2 versus controls [125, 126]. However, copy-number variation analyses showed that MPC2 was frequently amplified in a specific molecular subtype of HCC termed as the “mixed group” subtype characterized by higher variability in metabolic activity, compared to the quiescent subtype defined by low metabolic activity and cholesterol subtype characterized by reliance on cholesterol synthesis pathways [127]. Hence, direct mechanistic studies of TM4SF4 regulation of genes expression are required to demonstrate the downstream signaling controlled by TM4SF4 in HCC.

The figure legends of Fig. 8 have also been expanded to include this sentence:

The top eight genes with the highest r values with TM4SF4 expression were labeled in the scatter plot.

(2) Please explain why the authors showed the result in Fig.8? Although the structure of the inhibitory site of TM4SF4 protein is abruptly introduced in Fig. 8, I do not feel the significance of treating it in the text unless the inhibitors are actually synthesized and tested in mice or other animals.

Thank you for the comments. The molecular docking analyses were initially included to address the therapeutic aspect of the manuscript. Nonetheless, we have now completely removed the results of this figure (previously designated as Fig. 8), its methods, discussion, and all corresponding texts from the manuscript to maintain focus on the core findings of the study. We apologize for any confusion caused.

References

81. Liu J, Wang D, Zhang C, Zhang Z, Chen X, Lian J, et al. Identification of liver metastasis-associated genes in human colon carcinoma by mRNA profiling. Chin J Cancer Res. 2018;30(6):633-46. doi: 10.21147/j.issn.1000-9604.2018.06.08. PMID: 30700932.

83. Vandewynckel YP, Laukens D, Devisscher L, Paridaens A, Bogaerts E, Verhelst X, et al. Tauroursodeoxycholic acid dampens oncogenic apoptosis induced by endoplasmic reticulum stress during hepatocarcinogen exposure. Oncotarget. 2015;6(29):28011-25. doi: 10.18632/oncotarget.4377. PMID: 26293671.

84. Castro RE, Sola S, Ma X, Ramalho RM, Kren BT, Steer CJ, et al. A distinct microarray gene expression profile in primary rat hepatocytes incubated with ursodeoxycholic acid. J Hepatol. 2005;42(6):897-906. doi: 10.1016/j.jhep.2005.01.026. PMID: 15885361.

89. Tang Z, Kang B, Li C, Chen T, Zhang Z. GEPIA2: an enhanced web server for large-scale expression profiling and interactive analysis. Nucleic Acids Res. 2019;47(W1):W556-W60. doi: 10.1093/nar/gkz430. PMID: 31114875.

125. Wyler E, Mosbauer K, Franke V, Diag A, Gottula LT, Arsie R, et al. Transcriptomic profiling of SARS-CoV-2 infected human cell lines identifies HSP90 as target for COVID-19 therapy. iScience. 2021;24(3):102151. doi: 10.1016/j.isci.2021.102151. PMID: 33585804.

126. Vastrad BM, Vastrad CM. Bioinformatics analysis of expression profiling by high throughput sequencing for identification of potential key genes among SARS-CoV-2/COVID 19. 2021. doi: 10.21203/rs.3.rs-122015/v2.

127. Deng W, Zhu P, Xu H, Hou X, Chen W. Classification and Prognostic Characteristics of Hepatocellular Carcinoma Based on Glycolysis Cholesterol Synthesis Axis. J Oncol. 2022;2022:2014625. doi: 10.1155/2022/2014625. PMID: 36213830.

Reviewer #2

The authors of this article have applied a multi-omics data analysis of TM4SF4 as a cell surface target in hepatocellular carcinoma. This title is worthy of investigation and it is evident that lots of effort was put into writing this informative manuscript. However, some comments are mentioned below that need more clarification, explanation, or modification.

1. Please include the type of study in the title.

Thank you for the comment. The title has been revised to state the type of study as follows: **********

Multiomics in silico analysis identifies TM4SF4 as a cell surface target in hepatocellular carcinoma

2. The abstract of the article is better to be structured.

Thank you for the comment. The Abstract has been revised to be more structured while adhering to PLOS One's Abstract format as follows:

The clinical application of cellular immunotherapy in hepatocellular carcinoma (HCC) is impeded by the lack of a cell surface target frequently expressed in HCC cells and with minimal presence in normal tissues to reduce on-target, off-tumor toxicity. To address this, an in silico multomics analysis was conducted to identify an optimal therapeutic target in HCC. A longlist of genes (n=12,948) expressed in HCCs according to The Human Protein Atlas database were examined. Eight genes were shortlisted to identify one with the highest expression in HCCs, without being shed into circulation, and with restrictive expression profile in other normal human tissues. A total of eight genes were shortlisted and subsequently ranked according to the combination of their transcript and protein expression levels in HCC cases (n=791) derived from four independent datasets. TM4SF4 was the top-ranked target with the highest expression in HCCs. TM4SF4 showed more favorable expression profile with significantly lower expression in normal human tissues but more highly expressed in HCC compared with seven other common HCC therapeutic targets. Furthermore, scRNA-seq and immunohistoc

---

## [Decision Letter · Decision Letter 1]

21 Jan 2025

Multiomics in silico analysis identifies TM4SF4 as a cell surface target in hepatocellular carcinoma

PONE-D-24-26145R1

Dear Dr. Wong,

We’re pleased to inform you that your manuscript has been judged scientifically suitable for publication and will be formally accepted for publication once it meets all outstanding technical requirements.

Kind regards,

Dr Mahmood S Choudhery, PhD

Academic Editor

PLOS ONE

Additional Editor Comments (optional):

Reviewers' comments:

Reviewer's Responses to Questions

**Comments to the Author**

1. If the authors have adequately addressed your comments raised in a previous round of review and you feel that this manuscript is now acceptable for publication, you may indicate that here to bypass the “Comments to the Author” section, enter your conflict of interest statement in the “Confidential to Editor” section, and submit your "Accept" recommendation.

Reviewer #1: All comments have been addressed

2. Is the manuscript technically sound, and do the data support the conclusions?

Reviewer #1: Yes

3. Has the statistical analysis been performed appropriately and rigorously? 

Reviewer #1: I Don't Know

4. Have the authors made all data underlying the findings in their manuscript fully available?

Reviewer #1: Yes

5. Is the manuscript presented in an intelligible fashion and written in standard English?

Reviewer #1: Yes

6. Review Comments to the Author

Reviewer #1: The authors replied all of my comments clearly, and the readers will be more impressed in this manuscripts.

7. PLOS authors have the option to publish the peer review history of their article (what does this mean? ). If published, this will include your full peer review and any attached files.

**Do you want your identity to be public for this peer review?** For information about this choice, including consent withdrawal, please see our Privacy Policy .

Reviewer #1: **Yes: ** Sadahiro Iwabuchi

---

## [Editor Report · Acceptance letter]

PONE-D-24-26145R1

PLOS ONE

Dear Dr. Wong,

I'm pleased to inform you that your manuscript has been deemed suitable for publication in PLOS ONE. Congratulations! Your manuscript is now being handed over to our production team.

Kind regards,

on behalf of

Dr. Mahmood S Choudhery

Academic Editor

PLOS ONE